# SHORTEST-PATH CONSTRAINED REINFORCEMENT LEARNING FOR SPARSE REWARD TASKS

## ABSTRACT

We propose the $k$-Shortest-Path ($k$-SP) constraint: a novel constraint on the agent's trajectory that improves the sample-efficiency in sparse-reward MDPs. We show that any optimal policy necessarily satisfies the $k$-SP constraint. Notably, the $k$-SP constraint prevents the policy from exploring state-action pairs along the non-$k$-SP trajectories (*e.g.*, going back and forth). However, in practice, excluding state-action pairs may hinder convergence of RL algorithms. To overcome this, we propose a novel cost function that penalizes the policy violating SP constraint, instead of completely excluding it. Our numerical experiment in a tabular RL setting demonstrate that the SP constraint can significantly reduce the trajectory space of policy. As a result, our constraint enables more sample efficient learning by suppressing redundant exploration and exploitation. Our experiments on *MiniGrid* and *DeepMind Lab* show that the proposed method significantly improves proximal policy optimization (PPO) and outperforms existing novelty-seeking exploration methods including count-based exploration, indicating that it improves the sample efficiency by preventing the agent from taking redundant actions.

## 1 INTRODUCTION

Recently, deep reinforcement learning (RL) has achieved a large number of breakthroughs in many domains including video games (Mnih et al., 2015; Vinyals et al., 2019), and board games (Silver et al., 2017). Nonetheless, a central challenge in reinforcement learning (RL) is the sample efficiency (Kakade et al., 2003); it has been shown that the RL algorithm requires a large number of samples for successful learning in MDP with large state and action space. Moreover, the success of RL algorithm heavily hinges on the quality of collected samples; the RL algorithm tends to fail if the collected trajectory does not contain enough evaluative feedback (*e.g.*, sparse or delayed reward).

To circumvent this challenge, planning-based methods utilize the environment's model to improve or create a policy instead of interacting with environment. Recently, combining the planning method with an efficient path search algorithm, such as Monte-Carlo tree search (MCTS) (Norvig, 2002; Coulom, 2006), has demonstrated successful results (Guo et al., 2016; Vodopivec et al., 2017; Silver et al., 2017). However, such tree search methods would require an accurate model of MDP and the complexity of planning may grow intractably large for complex domain. Model-based RL methods attempt to learn a model instead of assuming that model is given, but learning an accurate model also requires a large number of samples, which is often even harder to achieve than solving the given task. Model-free RL methods can be learned solely from the environment reward, without the need of a (learned) model. However, both value-based and policy-based methods suffer from poor sample efficiency especially in sparse-reward tasks. To tackle sparse reward problems, researchers have proposed to learn an intrinsic bonus function that measures the novelty of the state that agent visits (Schmidhuber, 1991; Oudeyer & Kaplan, 2009; Pathak et al., 2017; Savinov et al., 2018b; Choi et al., 2018; Burda et al., 2018). However, when such intrinsic bonus is added to the reward, it often requires a careful balancing between environment reward and bonus and scheduling of the bonus scale in order to guarantee the convergence to optimal solution.

To tackle aforementioned challenge of sample efficiency in sparse reward tasks, we introduce a constrained-RL framework that improves the sample efficiency of any model-free RL algorithm in sparse-reward tasks, under the mild assumptions on MDP (see Appendix G). Of note, though our framework will be formulated for policy-based methods, our final form of cost function (Eq. (10) in Section 4) is applicable to both policy-based and value-based methods. We propose a novel *k-shortest-path* ($k$-SP) constraint (Definition 7) that improves sample efficiency of policy learning (See Figure 1). The $k$-SP constraint is applied to a trajectory rolled out by a policy; all of its sub-path

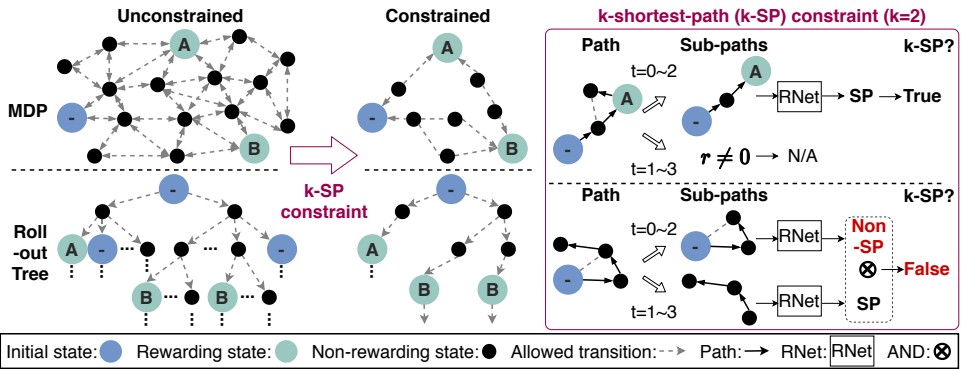

Figure 1: The $k$-SP constraint improves sample efficiency of RL methods in sparse-reward tasks by pruning out suboptimal trajectories from the trajectory space. Intuitively, the $k$-SP constraint means that when a policy rolls out into trajectories, all of sub-paths of length $k$ is a shortest path (under a distance metric defined in terms of policy, discount factor and transition probability; see Section 3.2 for the formal definition). **(Left)** MDP and a rollout tree are given. **(Middle)** The paths that satisfy the $k$-SP constraint. The number of admissible trajectories is drastically reduced. **(Right)** A path rolled out by a policy satisfies the $k$-SP constraint if all sub-paths of length $k$ are shortest paths and have not received non-zero reward. We use a reachability network to test if a given (sub-)path is a shortest path (See Section 4 for details).

of length $k$ is required to be a shortest-path under the $\pi$-*distance* metric which we define in Section 3.1. We prove that applying our constraint preserves the optimality for any MDP (Theorem 3), except the stochastic and multi-goal MDP which requires additional assumptions. We relax the hard constraint into a soft cost formulation (Tessler et al., 2019), and use a *reachability network* (Savinov et al., 2018b) (RNet) to efficiently learn the cost function in an off-policy manner.

We summarize our contributions as the following: **(1)** We propose a novel constraint that can improve the sample efficiency of any model-free RL method in sparse reward tasks. **(2)** We present several theoretical results including the proof that our proposed constraint preserves the optimal policy of given MDP. **(3)** We present a numerical result in tabular RL setting to precisely evaluate the effectiveness of the proposed method. **(4)** We propose a practical way to implement our proposed constraint, and demonstrate that it provides a significant improvement on two complex deep RL domains. **(5)** We demonstrate that our method significantly improves the sample-efficiency of PPO, and outperforms existing novelty-seeking methods on two complex domains in sparse reward setting.

## 2 PRELIMINARIES

**Markov Decision Process (MDP).** We model a task as an MDP tuple $\mathcal{M} = (\mathcal{S}, \mathcal{A}, \mathcal{P}, \mathcal{R}, \rho, \gamma)$, where $\mathcal{S}$ is a state set, $\mathcal{A}$ is an action set, $\mathcal{P}$ is a transition probability, $\mathcal{R}$ is a reward function, $\rho$ is an initial state distribution, and $\gamma \in [0, 1)$ is a discount factor. For each state $s$, the value of a policy $\pi$ is denoted by $V^\pi(s) = \mathbb{E}^\pi[\sum_t \gamma^t r_t \mid s_0 = s]$. Then, the goal is to find the optimal policy $\pi^*$ that maximizes the expected return:

$$\pi^* = \arg\max_\pi \mathbb{E}^\pi_{s \sim \rho}\Big[\sum_t \gamma^t r_t \mid s_0 = s\Big] = \arg\max_\pi \mathbb{E}_{s \sim \rho}[V^\pi(s)]. \tag{1}$$

**Constrained MDP.** A constrained Markov Decision Process (CMDP) is an MDP with extra constraints that restrict the domain of allowed policies (Altman, 1999). Specifically, CMDP introduces a constraint function $C(\pi)$ that maps a policy to a scalar, and a threshold $\alpha \in \mathbb{R}$. The objective of CMDP is to maximize the expected return $R(\tau) = \sum_t \gamma^t r_t$ of a trajectory $\tau = \{s_0, a_0, r_1, s_1, a_1, r_2, s_2, \ldots\}$ subject to a constraint: $\pi^* = \arg\max_\pi \mathbb{E}_{\tau \sim \pi}[R(\tau)]$, s.t. $C(\pi) \le \alpha$.

A popular choice of constraint is based on the transition cost function (Tessler et al., 2019) $c(s, a, r, s') \in \mathbb{R}$ which assigns a scalar-valued cost to each transition. Then the constraint function for a policy $\pi$ is defined as the discounted sum of the cost under the policy: $C(\pi) = \mathbb{E}_{\tau \sim \pi}\left[\sum_t \gamma^t c(s_t, a_t, r_{t+1}, s_{t+1})\right]$. In this work, we propose a *shortest-path* constraint, that provably preserves the optimal policy of the original unconstrained MDP, while reducing the trajectory space. We will use a cost function-based formulation to implement our constraint (see Section 3 and 4).

## 3 FORMULATION: $k$-SHORTEST PATH CONSTRAINT

We define the $k$-shortest-path (k-SP) constraint to remove redundant transitions (*e.g.*, unnecessarily going back and forth), leading to faster policy learning. We show two important properties of our constraint: (1) the optimal policy is preserved, and (2) the policy search space is reduced.

In this work, we limit our focus to MDPs satisfying $R(s) + \gamma V^*(s) > 0$ for all initial states $s \in \rho$ and all rewarding states that optimal policy visits with non-zero probability $s \in \{s | r(s) \neq 0, \pi^*(s) > 0\}$. We exploit this mild assumption to prove that our constraint preserves optimality. Intuitively, we exclude the case when the optimal strategy for the agent is at best choosing a "lesser of evils" (*i.e.*, largest but negative value) which often still means a failure. We note that this is often caused by unnatural reward function design; in principle, we can avoid this by simply offsetting reward function by a constant $-|\min_{s \in \{s | \pi^*(s) > 0\}} V^*(s)|$ for every transition, assuming the policy is *proper*[1]. Goal-conditioned RL (Nachum et al., 2018) and most of the well-known domains such as *Atari* (Bellemare et al., 2013), *DeepMind Lab* (Beattie et al., 2016), *MiniGrid* (Chevalier-Boisvert et al., 2018), etc., satisfy this assumption. Also, for general settings with stochastic MDP and multi-goals, we require additional assumptions to prove the optimality guarantee (See Appendix G for details).

### 3.1 SHORTEST-PATH POLICY AND SHORTEST-PATH CONSTRAINT

Let $\tau$ be a *path* defined by a sequence of states: $\tau = \{s_0, \ldots, s_{\ell(\tau)}\}$, where $\ell(\tau)$ is the *length* of a path $\tau$ (*i.e.*, $\ell(\tau) = |\tau| - 1$). We denote the set of all paths from $s$ to $s'$ by $\mathcal{T}_{s,s'}$. A path $\tau^*$ from $s$ to $s'$ is called a *shortest path* from $s$ to $s'$ if $\ell(\tau)$ is minimum, *i.e.*, $\ell(\tau^*) = \min_{\tau \in \mathcal{T}_{s,s'}} \ell(\tau)$.

Now we will define similar concepts (length, shortest path, *etc.*) *with respect to* a policy. Intuitively, a policy that rolls out shortest paths (up to some stochasticity) to a goal state or between any state pairs should be a counterpart. We consider a set of all admissible paths from $s$ to $s'$ under a policy $\pi$:

**Definition 1** (Path set). $\mathcal{T}_{s,s'}^{\pi} = \{\tau \mid s_0 = s, s_{\ell(\tau)} = s', p_\pi(\tau) > 0, \{s_t\}_{t < \ell(\tau)} \neq s'\}$. *That is, $\mathcal{T}_{s,s'}^{\pi}$ is a set of all paths that policy $\pi$ may roll out from $s$ and terminate once visiting $s'$.*

If the MDP is a single-goal task, *i.e.*, there exists a unique (rewarding) goal state $s_g \in \mathcal{S}$ such that $s_g$ is a terminal state, and $R(s) > 0$ if and only if $s = s_g$, any shortest path from an initial state to the goal state is the optimal path with the highest return $\tilde{R}(\tau)$, and a policy that rolls out a shortest path is therefore optimal (see Lemma 4).[2] This is because all states except for $s_g$ are non-rewarding states, but in general MDPs this is not necessarily true. However, this motivates us to limit the domain of shortest path to among *non-rewarding states*. We define *non-rewarding paths* from $s$ to $s'$ as follows:

**Definition 2** (Non-rewarding path set). $\mathcal{T}_{s,s',\mathrm{nr}}^{\pi} = \{\tau \mid \tau \in \mathcal{T}_{s,s'}^{\pi}, \{r_t\}_{t < \ell(\tau)} = 0\}$.

In words, $\mathcal{T}_{s,s',\mathrm{nr}}^{\pi}$ is a set of all non-rewarding paths from $s$ to $s'$ rolled out by policy $\pi$ (*i.e.*, $\tau \in \mathcal{T}_{s,s'}^{\pi}$) without any associated reward except the last step (*i.e.*, $\{r_t\}_{t < |\tau|} = 0$). Now we are ready to define a notion of length with respect to a policy and shortest path policy:

**Definition 3** ($\pi$-distance from $s$ to $s'$). $D_{\mathrm{nr}}^{\pi}(s, s') = \log_\gamma \left( \mathbb{E}_{\tau \sim \pi: \tau \in \mathcal{T}_{s,s',\mathrm{nr}}^{\pi}} \left[ \gamma^{\ell(\tau)} \right] \right)$

**Definition 4** (Shortest path distance from $s$ to $s'$). $D_{\mathrm{nr}}(s, s') = \min_\pi D_{\mathrm{nr}}^{\pi}(s, s')$.

We define $\pi$-distance to be the log-mean-exponential of the length $\ell(\tau)$ of non-rewarding paths $\tau \in \mathcal{T}_{s,s',\mathrm{nr}}^{\pi}$. When there exists no admissible path from $s$ to $s'$ under policy $\pi$, the path length is defined to be $\infty$: $D_{\mathrm{nr}}^{\pi}(s, s') = \infty$ if $\mathcal{T}_{s,s',\mathrm{nr}}^{\pi} = \emptyset$. We note that when both MDP and policy are deterministic, $D^{\pi}(s, s')$ recovers the natural definition of path length, $D_{\mathrm{nr}}^{\pi}(s, s') = \ell(\tau)$.

We call a policy a *shortest-path policy* from $s$ to $s'$ if it roll outs a path with the smallest $\pi$-distance:

**Definition 5** (Shortest path policy from $s$ to $s'$). $\pi \in \Pi_{s \to s'}^{SP} = \{\pi \in \Pi \mid D_{\mathrm{nr}}^{\pi}(s, s') = D_{\mathrm{nr}}(s, s')\}$.

Finally, we will define the shortest-path (SP) constraint. Let $\mathcal{S}^{\mathrm{IR}} = \{s \mid R(s) > 0 \text{ or } \rho(s) > 0\}$ be the union of all initial and rewarding states, and $\Phi^\pi = \{(s, s') \mid s, s' \in \mathcal{S}^{\mathrm{IR}}, \rho(s) > 0, \mathcal{T}_{s,s',\mathrm{nr}}^{\pi} \neq \emptyset\}$ be the subset of $\mathcal{S}^{\mathrm{IR}}$ such that agent may roll out. Then, the SP constraint is applied to the non-rewarding sub-paths between states in $\Phi^\pi$: $\mathcal{T}_{\Phi,\mathrm{nr}}^{\pi} = \bigcup_{(s,s') \in \Phi^\pi} \mathcal{T}_{s,s',\mathrm{nr}}^{\pi}$. We note that these definitions are used in the proofs (Appendix G). Now, we define the shortest-path constraint as follows:

**Definition 6** (Shortest-path constraint). *A policy $\pi$ satisfies the shortest-path (SP) constraint if $\pi \in \Pi^{SP}$, where $\Pi^{SP} = \{\pi \mid \text{For all } s, s' \in \mathcal{T}_{\Phi,\mathrm{nr}}^{\pi}, \text{it holds } \pi \in \Pi_{s \to s'}^{SP}\}$.*

---

[1]It is an instance of potential-based reward shaping which has optimality guarantee (Ng et al., 1999).

[2]We refer the readers to Appendix F for more detailed discussion and proofs for single-goal MDPs.

Intuitively, the SP constraint forces a policy to transition between initial and rewarding states via shortest paths. The SP constraint would be particularly effective in sparse-reward settings, where the distance between rewarding states is large.

Given these definitions, we can show that an optimal policy indeed satisfies the SP constraint in a general MDP setting. In other words, the shortest path constraint should not change optimality:

**Theorem 1.** *For any MDP, an optimal policy $\pi^*$ satisfies the shortest-path constraint: $\pi^* \in \Pi^{SP}$.*

*Proof.* See Appendix G for the proof. □

### 3.2 Relaxation: $k$-Shortest-Path Constraint

Implementing the shortest-path constraint is, however, intractable since it requires a distance predictor $D_{\mathrm{nr}}(s, s')$. Note that the distance predictor addresses the optimization problem, which might be as difficult as solving the given task. To circumvent this challenge, we consider its more tractable version, namely a $k$-*shortest path* constraint, which reduces the shortest-path problem $D_{\mathrm{nr}}(s, s')$ to a binary decision problem — is the state $s'$ reachable from $s$ within $k$ steps? — also known as $k$-reachability (Savinov et al., 2018b). The $k$-shortest path constraint is defined as follows:

**Definition 7** ($k$-shortest-path constraint). *A policy $\pi$ satisfies the $k$-shortest-path constraint if $\pi \in \Pi_k^{SP}$, where*

$$\Pi_k^{SP} = \{\pi \mid \textit{For all } s, s' \in \mathcal{T}_{\Phi,\mathrm{nr}}^{\pi}, D_{\mathrm{nr}}^{\pi}(s, s') \leq k, \textit{it holds } \pi \in \Pi_{s \to s'}^{SP}\}. \quad (2)$$

Note that the SP constraint (Definition 6) is relaxed by adding a condition $D_{\mathrm{nr}}^{\pi}(s, s') \leq k$. In other words, the $k$-SP constraint is imposed only for $s, s'$-path whose length is not greater than $k$. From Eq. (2), we can prove an important property and then Theorem 3 (optimality):

**Lemma 2.** *For an MDP $\mathcal{M}$, $\Pi_m^{SP} \subset \Pi_k^{SP}$ if $k < m$.*

*Proof.* It is true since $\{(s, s') \mid D_{\mathrm{nr}}^{\pi}(s, s') \leq k\} \subset \{(s, s') \mid D_{\mathrm{nr}}^{\pi}(s, s') \leq m\}$ for $k < m$. □

**Theorem 3.** *For an MDP $\mathcal{M}$ and any $k \in \mathbb{R}$, an optimal policy $\pi^*$ is a $k$-shortest-path policy.*

*Proof.* Theorem 1 tells $\pi^* \in \Pi^{SP}$. Eq. (2) tells $\Pi^{SP} = \Pi_\infty^{SP}$ and Lemma 2 tells $\Pi_\infty^{SP} \subset \Pi_k^{SP}$. Collectively, we have $\pi^* \in \Pi^{SP} = \Pi_\infty^{SP} \subset \Pi_k^{SP}$. □

In conclusion, Theorem 3 states that the $k$-SP constraint does not change the optimality of policy, and Lemma 2 states a larger $k$ results in a larger reduction in policy search space. Thus, it motivates us to apply the $k$-SP constraint in policy search to more efficiently find an optimal policy. For the numerical experiment on measuring the reduction in the policy roll-outs space, please refer to Section 6.4.

## 4 SPRL: Shortest-Path Reinforcement Learning

$k$**-Shortest-Path Cost.** The objective of RL with the $k$-SP constraint $\Pi_k^{\mathrm{SP}}$ can be written as:

$$\pi^* = \arg\max_\pi \mathbb{E}^\pi [R(\tau)], \quad \text{s.t. } \pi \in \Pi_k^{\mathrm{SP}}, \quad (3)$$

where $\Pi_k^{\mathrm{SP}} = \{\pi \mid \forall (s, s' \in \mathcal{T}_{\Phi,\mathrm{nr}}^{\pi}), D_{\mathrm{nr}}^{\pi}(s, s') \leq k, \text{it holds } \pi \in \Pi_{s \to s'}^{\mathrm{SP}}\}$ (Definition 7). We want to formulate the constraint $\pi \in \Pi_k^{\mathrm{SP}}$ in the form of constrained MDP (Section 2), *i.e.*, as $C(\pi) \leq \alpha$. We begin by re-writing the $k$-SP constraint into a cost-based form:

$$\Pi_k^{\mathrm{SP}} = \{\pi \mid C_k^{\mathrm{SP}}(\pi) = 0\}, \text{ where } C_k^{\mathrm{SP}}(\pi) = \sum_{(s, s' \in \mathcal{T}_{\Phi,\mathrm{nr}}^{\pi}): D_{\mathrm{nr}}^{\pi}(s, s') \leq k} \mathbb{I}[D_{\mathrm{nr}}(s, s') < D_{\mathrm{nr}}^{\pi}(s, s')]. \quad (4)$$

Note that $\mathbb{I}[D_{\mathrm{nr}}(s, s') < D_{\mathrm{nr}}^{\pi}(s, s')] = 0 \leftrightarrow D_{\mathrm{nr}}(s, s') = D_{\mathrm{nr}}^{\pi}(s, s')$ since $D_{\mathrm{nr}}(s, s') \leq D_{\mathrm{nr}}^{\pi}(s, s')$ from Definition 4. Similar to Tessler et al. (2019), we apply the constraint to the on-policy trajectory $\tau = (s_0, s_1, \dots)$ with discounting by replacing $(s, s')$ with $(s_t, s_{t+l})$ where $[t, t+l]$ represents each segment of $\tau$ with length $l$:

$$C_k^{\mathrm{SP}}(\pi) \simeq \mathbb{E}_{\tau \sim \pi}\left[\sum_{(t,l): t \geq 0, l \leq k} \gamma^t \cdot \mathbb{I}[D_{\mathrm{nr}}(s_t, s_{t+l}) < D_{\mathrm{nr}}^{\pi}(s_t, s_{t+l})] \cdot \mathbb{I}[\{r_j\}_{j=t}^{t+l-1} = 0]\right]$$

$$= \mathbb{E}_{\tau \sim \pi}\left[\sum_{(t,l): t \geq 0, l \leq k} \gamma^t \cdot \mathbb{I}\left[D_{\mathrm{nr}}(s_t, s_{t+l}) < \log_\gamma\left(\mathbb{E}_{\tau \in \mathcal{T}_{s_t, s_{t+l}, \mathrm{nr}}^{\pi}}[\gamma^{|\tau|}]\right)\right] \cdot \mathbb{I}[\{r_j\}_{j=t}^{t+l-1} = 0]\right]$$

$$\leq \mathbb{E}_{\tau \sim \pi}\left[\sum_{(t,l): t \geq 0, l \leq k} \gamma^t \cdot \mathbb{I}[D_{\mathrm{nr}}(s_t, s_{t+l}) < k] \cdot \mathbb{I}[\{r_j\}_{j=t}^{t+l-1} = 0]\right] \triangleq \widehat{C}_k^{\mathrm{SP}}(\pi). \quad (5)$$

---

**Algorithm 1** Reinforcement Learning with $k$-SP constraint (SPRL)

---

**Require:** Hyperparameters: $k \in \mathbb{N}, \lambda > 0$
1: **for** $n = 1, \ldots, N_{\text{policy}}$ **do**
2:      Rollout transitions $\tau = \{s_t, a_t, r_t\}_{t=1}^{|\tau|} \sim \pi$
3:      Compute the cost term $\{c_t\}_{t=1}^{|\tau|}$ as Eq. (13).
4:      Update policy $\pi$ to maximize the objective as Eq. (10) (e.g., run PPO train steps).
5:      Update Rnet training-buffer $\mathcal{B} = \mathcal{B} \cup \{\tau\}$.
6:      **if** $n \% T_{\text{Rnet}} = 0$ **then**           ▷ Periodically train Rnet for $N_{\text{Rnet}}$ times
7:          **for** $m = 1, \ldots, N_{\text{Rnet}}$ **do**
8:              Sample triplet $(s_{\text{anc}}, s_+, s_-) \sim \mathcal{B}$.        ▷ See Appendix D.3 for detail
9:              Update Rnet to minimize $\mathcal{L}_{\text{Rnet}}$ as Eq. (14).

---

Note that it is sufficient to consider only the cases $l = k$ (because for $l < k$, given $D_{\text{nr}}(s_t, s_{t+k}) < k$, we have $D(s_t, s_{t+l}) \leq l < k$). Then, we simplify $\widehat{C}_k^{\text{SP}}(\pi)$ as

$$\widehat{C}_k^{\text{SP}}(\pi) = \mathbb{E}_{\tau \sim \pi} \left[ \sum_t \gamma^t \mathbb{I}\left[ D_{\text{nr}}(s_t, s_{t+k}) < k \right] \cdot \mathbb{I}\left[ \{r_j\}_{j=t}^{t+k-1} = 0 \right] \right] \tag{6}$$

$$= \mathbb{E}_{\tau \sim \pi} \left[ \sum_t \gamma^t \mathbb{I}[t \geq k] \cdot \mathbb{I}\left[ D_{\text{nr}}(s_{t-k}, s_t) < k \right] \cdot \mathbb{I}\left[ \{r_j\}_{j=t-k}^{t-1} = 0 \right] \right]. \tag{7}$$

Finally, the per-time step cost $c_t$ is given as:

$$c_t = \mathbb{I}[t \geq k] \cdot \mathbb{I}\left[ D_{\text{nr}}(s_{t-k}, s_t) < k \right] \cdot \mathbb{I}\left[ \{r_j\}_{j=t-k}^{t-1} = 0 \right], \tag{8}$$

where $\widehat{C}_k^{\text{SP}}(\pi) = \mathbb{E}_{\tau \sim \pi}\left[ \sum_t \gamma^t c_t \right]$. Note that $\widehat{C}_k^{\text{SP}}(\pi)$ is an upper bound of $C_k^{\text{SP}}(\pi)$, which will be minimized by the bound to make as little violation of the shortest-path constraint as possible. Intuitively speaking, $c_t$ penalizes the agent from taking a non-$k$-shortest path at each step, so minimizing such penalties will make the policy satisfy the $k$-shortest-path constraint. In Eq. (8), $c_t$ depends on the previous $k$ steps; hence, the resulting CMDP becomes a $(k + 1)$-th order MDP. In practice, however, we empirically found that feeding only the current time-step observation to the policy performs better than stacking the previous $k$-steps of observations (See Appendix A.3 for details). Thus, we did not stack the observation in all the experiments. We use the Lagrange multiplier method to convert the objective (3) into an equivalent unconstrained problem as follows:

$$\min_{\lambda > 0} \max_{\theta} L(\lambda, \theta) = \min_{\lambda > 0} \max_{\theta} \mathbb{E}_{\tau \sim \pi_\theta} \left[ \sum_t \gamma^t \left( r_t - \lambda c_t \right) \right], \tag{9}$$

where $L$ is the Lagrangian, $\theta$ is the parameter of policy $\pi$, and $\lambda > 0$ is the Lagrangian multiplier. Since Theorem 3 shows that the shortest-path constraint preserves the optimality, we are free to set any $\lambda > 0$. Thus, we simply consider $\lambda$ as a tunable *positive* hyperparameter, and simplify the min-max problem (9) to an RL objective with costs $c_t$ being added:

$$\max_{\theta} \mathbb{E}_{\tau \sim \pi_\theta} \left[ \sum_t \gamma^t \left( r_t - \lambda c_t \right) \right]. \tag{10}$$

**Practical implementation of the cost function.** We implement the binary distance discriminator $\mathbb{I}(D_{\text{nr}}(s_{t-k}, s_t) < k)$ in Eq. (8) using *k-reachability network* (Savinov et al., 2018b). The $k$-reachability network $\text{Rnet}_k(s, s')$ is trained to output 1 if the state $s'$ is reachable from the state $s$ with less than or equal to $k$ consecutive actions, and 0 otherwise. Formally, we take the functional form: $\text{Rnet}_k(s, s') \simeq \mathbb{I}(D_{\text{nr}}(s, s') < k + 1)$. We then estimate the cost term $c_t$ using $(k - 1)$-reachability network as follows:

$$c_t = \mathbb{I}\left[ D_{\text{nr}}(s_{t-k}, s_t) < k \right] \cdot \mathbb{I}\left[ \{r_l\}_{l=t-k}^{t-1} = 0 \right] \cdot \mathbb{I}[t \geq k] \tag{11}$$

$$= \text{Rnet}_{k-1}(s_{t-k}, s_t) \cdot \mathbb{I}\left[ \{r_l\}_{l=t-k}^{t-1} = 0 \right] \cdot \mathbb{I}[t \geq k]. \tag{12}$$

Intuitively speaking, if the agent takes a $k$-shortest path, then the distance between $s_{t-k}$ and $s_t$ is $k$, hence $c_t = 0$. If it is not a $k$-shortest path, $c_t > 0$ since the distance between $s_{t-k}$ and $s_t$ will be less than $k$. In practice, due to the error in the reachability network, we add a small tolerance $\Delta t \in \mathbb{N}$ to ignore outliers. It leads to an empirical version of the cost as follows:

$$c_t \simeq \text{Rnet}_{k-1}(s_{t-k-\Delta t}, s_t) \cdot \mathbb{I}\left[ \{r_l\}_{l=t-k-\Delta t}^{t-1} = 0 \right] \cdot \mathbb{I}(t \geq k + \Delta t). \tag{13}$$

In our experiment, we found that a small tolerance $\Delta t \simeq k/5$ works well in general. Similar to Savinov et al. (2018b), we used the following contrastive loss for training the reachability network:

$$\mathcal{L}_{\text{Rnet}} = -\log\left(\text{Rnet}_{k-1}(s_{\text{anc}}, s_+)\right) - \log\left(1 - \text{Rnet}_{k-1}(s_{\text{anc}}, s_-)\right), \tag{14}$$

where $s_{\text{anc}}, s_+, s_-$ are the anchor, positive, and negative samples, respectively (See Appendix D.3 for the detail of training).

## 5 RELATED WORK

**Connection between shortest-path problem and planning.** Many early works (Bellman, 1958; Ford Jr, 1956; Bertsekas & Tsitsiklis, 1991; 1995) have discussed (stochastic) shortest path problems in the context of MDP. They viewed the shortest-path problem as planning problem and proposed a dynamic programming-based algorithm similar to the value iteration (Sutton & Barto, 2018) to solve it. Our main idea is inspired by (but not based on) this viewpoint. Specifically, our method does not directly solve the shortest path problem via planning; hence, our method does not require model. Our method only exploits the optimality guarantee of the shortest-path under the $\pi$-distance to prune out sub-optimal policies (*i.e.*, non-shortest paths).

**Distance metric in goal-conditioned RL.** In goal-conditioned RL, there has been a recent surge of interest on learning a distance metric in state (or goal) space for to construct a high-level MDP graph and perform planning to find a shortest-path to the goal state. Huang et al. (2019); Laskin et al. (2020) used the universal value function (UVF) (Schaul et al., 2015) with a constant step penalty as a distance function. Zhang et al. (2018); Laskin et al. (2020) used the success rate of transition between nodes as distance and searched for the longest path to find the plan with highest success rate. SPTM (Savinov et al., 2018a) defined a binary distance based on reachability network (RNet) to connect near by nodes in the graph. However, the proposed distance metrics and methods can be used only for the goal-conditioned task and lacks the theoretical guarantee in general MDP, while our theory and framework are applicable to general MDP (see Section 3.1).

**Reachability network.** The reachability network (RNet) was first proposed by Savinov et al. (2018b) as a way to measure the novelty of a state for *exploration*. Intuitively, if current state is not reachable from previous states in episodic memory, it is considered to be novel. SPTM (Savinov et al., 2018a) used RNet to predict the local connectivity (*i.e.*, binary distance) between observations in memory for *graph-based planning* in navigation task. On the other hand, we use RNet for *constraining the policy* (*i.e.*, removing the sub-optimal policies from policy space). Thus, in ours and in other two compared works, RNet is being employed for fundamentally different purposes.

**More related works.** Please refer to Appendix H for further discussions about other related works.

## 6 EXPERIMENTS

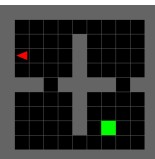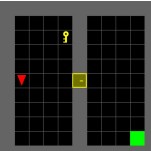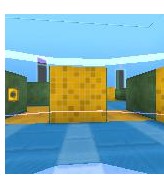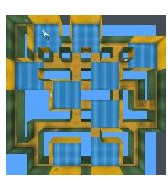

Figure 2: An example observation of (a) *FourRooms-11×11*, (b) *KeyDoors-11×11* in *MiniGrid*, (c) *GoalLarge* in *DeepMind Lab*, and (d) the maze layout (not available to the agent) of *GoalLarge*.

### 6.1 SETTINGS

**Environments.** We evaluate our **SPRL** on two challenging domains: *MiniGrid* (Chevalier-Boisvert et al., 2018) and *DeepMind Lab* (Beattie et al., 2016). *MiniGrid* is a 2D grid world environment with challenging features such as pictorial observation, random initialization of the agent and the goal, complex state and action space where coordinates, directions, and other object statuses (*e.g.*, key-door) are considered. We conducted experiments on four standard tasks: *FourRooms-7×7*, *FourRooms-11×11*, *KeyDoors-7×7*, and *KeyDoors-11×11*. *DeepMind Lab* is a 3D environment with first person view. Along with the nature of partially-observed MDP, at each episode, the agent's initial and the goal location are reset randomly with a change of texture, maze structure, and colors. We conducted experiments on three standard tasks: *GoalSmall*, *GoalLarge*[3], and *ObjectMany*. We refer the readers to Figure 2 for examples of observations.

---

[3]*GoalLarge* task corresponds to the *Sparse* task in Savinov et al. (2018b), and our Figure 4 reproduces the result reported in Savinov et al. (2018b).

**Baselines.** We compared our methods with four baselines: **PPO** (Schulman et al., 2017), *episodic curiosity* (**ECO**) (Savinov et al., 2018b), *intrinsic curiosity module* (**ICM**) (Pathak et al., 2017), and **GT-Grid** (Savinov et al., 2018b). The **PPO** is used as a baseline RL algorithm for all other agents. The **ECO** agent is rewarded when it visits a state that is not reachable from the states in episodic memory within a certain number of actions; thus the novelty is only measured within an episode. Following Savinov et al. (2018b), we trained RNet in an off-policy manner from the agent's experience and used it for our **SPRL** and **ECO** on both *MiniGrid* (Section 6.2) and *DeepMind Lab* (Section 6.3). For the accuracy of the learned RNet on each task, please refer to the Appendix B. The **GT-Grid** agent has access to the agent's $(x, y)$ coordinates. It uniformly divides the world in 2D grid cells, and the agent is rewarded for visiting a novel grid cell. The **ICM** agent learns a forward and inverse dynamics model and uses the prediction error of the forward model to measure the novelty. We used the publicly available codebase (Savinov et al., 2018b) to obtain the baseline results. We used the same hyperparameter for all the tasks for a given domain — the details are described in the Appendix. We used the standard domain and tasks for reproducibility.

## 6.2 RESULTS ON *MiniGrid*

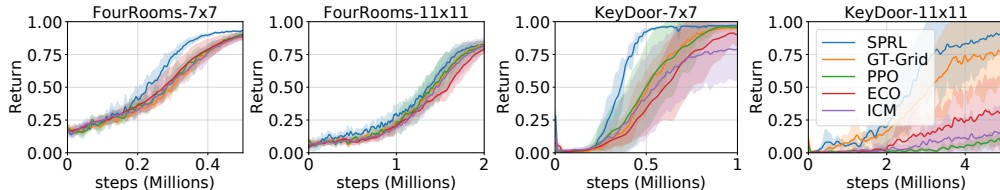

Figure 3: Progress of average episode reward on *MiniGrid* tasks. We report the mean (solid curve) and standard error (shadowed area) of the performance over six random seeds.

Figure 3 shows the performance of compared methods on *MiniGrid* domain. **SPRL** consistently outperforms all baseline methods over all tasks. We observe that exploration-based methods (*i.e.*, **ECO**, **ICM**, and **GT-Grid**) perform similarly to the **PPO** in the tasks with small state space (*e.g.*, *FourRooms-7×7* and *KeyDoors-7×7*). However, **SPRL** demonstrates a significant performance gain since it improves the exploitation by avoiding sub-optimality caused by taking a non-shortest-path.

## 6.3 RESULTS ON *DeepMind Lab*

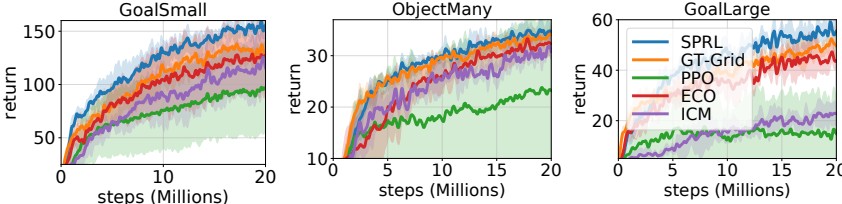

Figure 4: Progress of average episode reward on *DeepMind Lab* tasks. We report the mean (solid curve) and standard error (shadowed area) of the performance over four random seeds.

Figure 4 summarizes the performance of all the methods on *DeepMind Lab* tasks. Overall, our **SPRL** method achieves superior results compared to other methods. By the design of the task, the difficulty of exploration in each task increases in the order of *GoalSmall*, *ObjectMany*, and *GoalLarge* tasks, and we observe a coherent trend in the result. For harder exploration tasks, the exploration-based methods (**GT-Grid**, **ICM** and **ECO**) achieve a larger improvement over **PPO**: *e.g.*, 20%, 50%, and 100% improvement in *GoalSmall*, *ObjectMany*, and *GoalLarge*, respectively. As shown in Lemma 2, our **SPRL** is expected to have larger improvement for larger trajectory space (or state and action space) and sparser reward settings. We can verify this from the result: **SPRL** has the largest improvement in *GoalLarge* task, where both the map layout is largest and the reward is most sparse. Interestingly, **SPRL** even outperforms **GT-Grid** which simulates the upper-bound performance of novelty-seeking exploration method. This is possible since **SPRL** improves the exploration by suppressing unnecessary explorations, which is different from novelty-seeking methods, and also improves the exploitation by reducing the policy search space.

## 6.4 ANALYSIS ON $k$-SHORTEST-PATH CONSTRAINT

In this section, we numerically evaluate the effect of our $k$-shortest path constraint in tabular-RL setting. Specifically, we study the following questions: (1) Does the $k$-SP constraint with larger $k$

results in more reduction in trajectory space? (*i.e.*, validation of Lemma 2) (2) How much reduction in trajectory space does $k$-SP constraint provide with different $k$ and tolerance $\Delta t$?

**Experimental setup.** We implemented a simple tabular 7×7 Four-rooms domain where each state maps to a unique $(x, y)$ location of the agent. The agent can take *up, down, left, right* primitive actions to move to the neighboring state, and the episode horizon is set to 14 steps. The goal of the agent is reaching to the goal state, which gives +1 reward and terminates the episode. We computed the ground-truth distance between a pair of states to implement the $k$-shortest path constraint. We used the ground-truth distance function instead of the learned RNet to implement the exact SPRL agent.

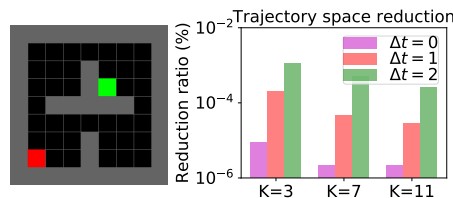

Figure 5: (Left) 7×7 Tabular four-rooms domain with initial agent location (red) and the goal location (green). (Right) The trajectory space reduction ratio (%) before and after constraining the trajectory space for various $k$ and $\Delta t$ with $k$-SP constraint. Even a small $k$ can greatly reduce the trajectory space with a reasonable tolerance $\Delta t$.

**Results.** Figure 5 summarizes the reduction in the trajectory space size. We searched over all possible trajectories of length 14 using breadth-first-search (BFS). Then we counted the number trajectories satisfying our $k$-SP constraint with varying parameters $k$ and tolerance $\Delta t$ and divided by total number of trajectories (*i.e.*, $4^{14} = 268M$). The result shows that our $k$-SP constraint drastically reduces the trajectory space even in a simple 2D grid domain; with very small $k = 3$ and no tolerance $\Delta t = 2$, we get only 24/268M size of the original search space. As we increase $k$, we can see more reduction in the trajectory space, which is consistent with Lemma 2. Also, increasing the tolerance $\Delta t$ slightly hurts the performance, but still achieves a large reduction (See Appendix A for more analysis on effect of $k$ and tolerance).

## 6.5 QUALITATIVE RESULT ON *MiniGrid*

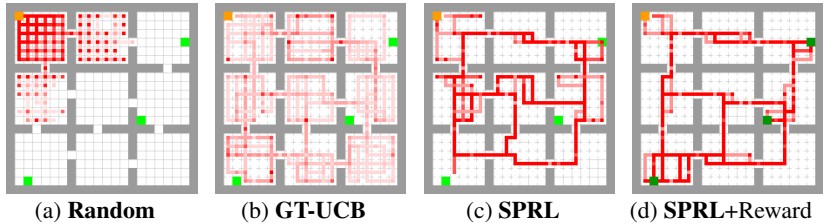

| (a) **Random** | (b) **GT-UCB** | (c) **SPRL** | (d) **SPRL+Reward** |

Figure 6: Transition count maps for baselines and **SPRL**: (a), (b), and (c) are in *reward-free* (light green) while (d) is in *reward-aware* (dark green) setting. In reward-free settings (a-c), we show rewarding states in light green only for demonstration purpose. The location of agent's initial state (orange) and rewarding states (dark green) are fixed. The episode length is limited to 500 steps.

We qualitatively study what type of policy is learned with the $k$-SP constraint with the ground-truth RNet in *NineRooms* domain of *MiniGrid*. Figure 6 (a-c) shows the converged behavior of our **SPRL** ($k = 15$), the ground-truth count-based exploration (Lai & Robbins, 1985) agent (**GT-UCB**) and uniformly random policy (**Random**) in a reward-free setting. We counted all the state transitions ($s_t \rightarrow s_{t+1}$) of each agent's roll-out and averaged over 4 random seeds. **Random** relies on the random walk and cannot explore further than the initial few rooms. **GT-UCB** seeks for a novel states, and visits all the states uniformly. **SPRL** learns to take a longest possible shortest path, which results in a "straight" path across the rooms. Note that this only represents a partial behavior of **SPRL**, since our cost also considers the *existence of non-zero reward* (see Eq. (8)). Thus, in (d), we tested **SPRL** while providing only the *existence* of non-zero reward (but not the reward magnitude). **SPRL** learns to take a shortest path between rewarding and initial states that is consistent with the shortest-path definition in Definition 7.

## 7 CONCLUSION

We presented the $k$-shortest-path constraint, which can improve the sample-efficiency of any model-free RL method by preventing the agent from taking a sub-optimal transition. We empirically showed that our **SPRL** outperforms vanilla RL and strong novelty-seeking exploration baselines on two challenging domains. We believe that our framework develops a unique direction for improving the sample efficiency in reinforcement learning; hence, combining our work with other techniques for better sample efficiency will be interesting future work that could benefit many practical tasks.

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

# Appendix: Shortest-path reinforcement learning

## A    MORE ABLATION STUDY

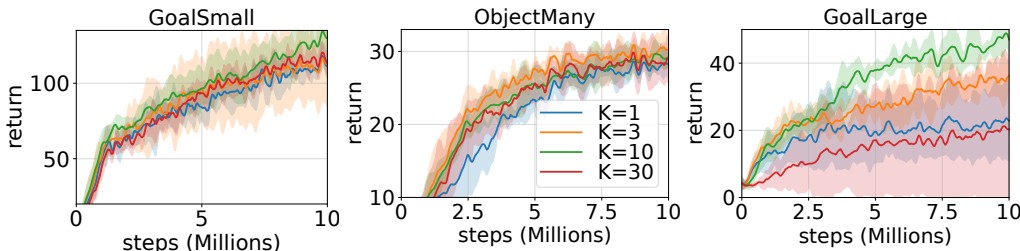

Figure 7: Average episode reward of **SPRL** with varying $k =$ 1, 3, 10, 30 as a function of environment steps for *DeepMind Lab* tasks. Other hyper-parameters are kept same as the best hyper-parameter. The best performance is obtained with $k = 10$.

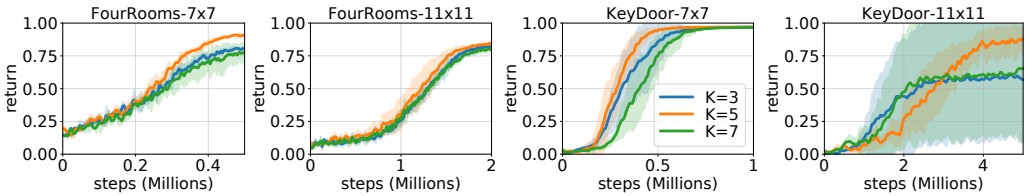

Figure 8: Average episode reward of **SPRL** with varying $k =$ 3, 5, 7 as a function of environment steps for *MiniGrid* tasks. Other hyper-parameters are kept same as the best hyper-parameter. The best performance is obtained with $k = 5$.

### A.1    EFFECT OF $k$

As proven in **Lemma** 2 and shown in Section 6.4, the larger $k$, the $k$-shortest constraint promises a larger reduction in policy space, which results in a faster learning. However, with our practical implementation of **SPRL** with a learned (imperfect) reachability network, overly large $k$ has a drawback. Intuitively speaking, it is harder for policy to satisfy the $k$-shortest constraint, and the supervision signal given by our cost function becomes sparser (*i.e.*, almost always penalized). Figure 7 and 8 shows the performance of **SPRL** on *DeepMind Lab* and *MiniGrid* domains with varying $k$. In both domains we can see that there exists a "sweet spot" that balances between the reduction in policy space and sparsity of the supervision (*e.g.*, $k = 10$ for *DeepMind Lab* and $k = 5$ for *MiniGrid*).

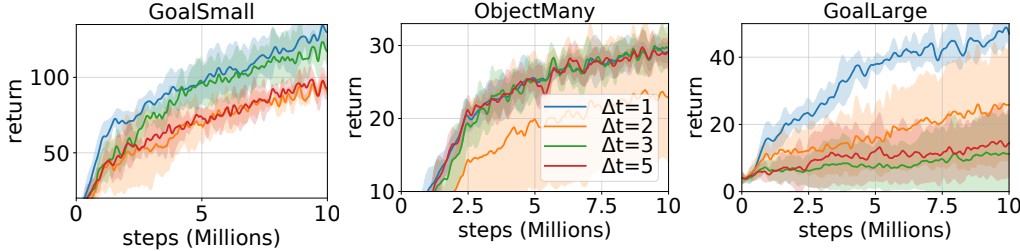

Figure 9: Average episode reward of **SPRL** with varying $\Delta t =$ 1, 2, 3, 5 as a function of environment steps for *DeepMind Lab* tasks. Other hyper-parameters are kept same as the best hyper-parameter. The best performance is obtained with $\Delta t = 1$.

### A.2    EFFECT OF TOLERANCE $\Delta t$

Adding the tolerance $\Delta t$ to our $k$-SP constraint makes it "softer" by allowing $\Delta t$-steps of redundancy in transition (See Eq. (13)). Intuitively, a small tolerance may improve the stability of RNet by incorporating a possible noise in RNet prediction, but too large tolerance will make it less effective on removing sub-optimality in transition. Figure 9 and 10 show the performance of **SPRL** on *DeepMind Lab* and *MiniGrid* domains with varying tolerance $\Delta t$. Similar to $k$, we can see that there exists a

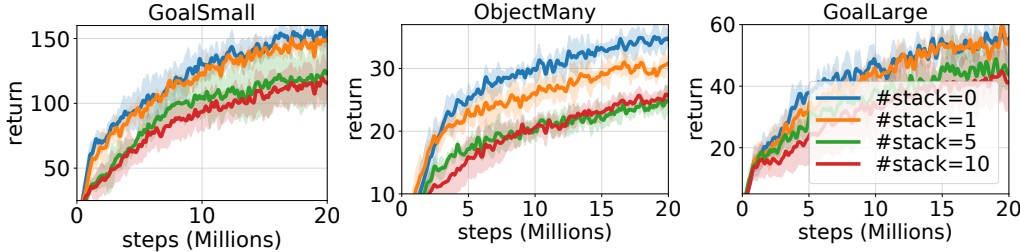

Figure 10: Average episode reward of **SPRL** with varying $\Delta t$ =10, 15, 25, 50 as a function of environment steps for *MiniGrid* tasks. Other hyper-parameters are kept same as the best hyper-parameter. The best performance is obtained with $\Delta t = 25$.

"sweet spot" that balances between the reduction in policy space and stabilization of noisy RNet output (*e.g.*, $\Delta t = 25$) in *MiniGrid*. Note that the best tolerance values for *DeepMind Lab* and *MiniGrid* are vastly different. This is mainly because we used *multiple tolerance sampling* (See Appendix D) for *DeepMind Lab* but not for *MiniGrid*. Since the multiple tolerance sampling also improves the stability of RNet, larger tolerance has less benefit compared to its disadvantage.

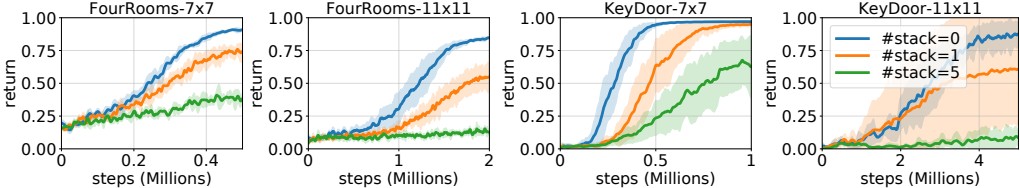

Figure 11: Average episode reward of **SPRL** with varying observation stacking dimension of 0, 1, 5, 10 as a function of environment steps for *DeepMind Lab* tasks. Other hyper-parameters are kept same as the best hyper-parameter. The best performance is obtained without stacking (*i.e.*, #stack=0).

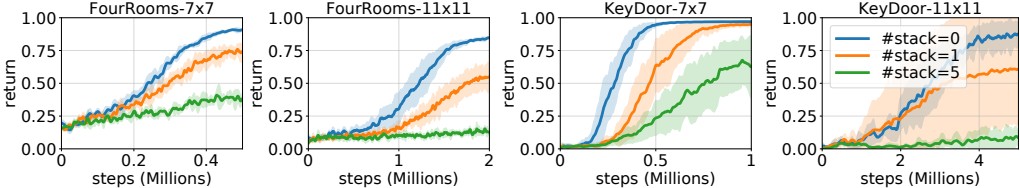

Figure 12: Average episode reward of **SPRL** with varying observation stacking dimension of 0, 1, 5 as a function of environment steps for *MiniGrid* tasks. Other hyper-parameters are kept same as the best hyper-parameter. The best performance is obtained without stacking (*i.e.*, #stack=0)

### A.3 STACKING OBSERVATION

The CMDP with $k$-SP constraint becomes the $(k+1)$-th order MDP as shown in Eq. (8). Thus, in theory, the policy should take currrent state $s_t$ augmented by stacking the $k$ previous states as input: $[s_{t-k}, s_{t-k+1} \ldots, s_t]$, where $[\cdot]$ is a stacks the pixel observation along the channel (*i.e.*, color) dimension. However, stacking the observation may not lead to the best empirical results in practice. Figure 11 and 12 show the performance of **SPRL** on *DeepMind Lab* and *MiniGrid* domains with varying stacking dimension. For stack=$m$, we stacked the observation from $t - m$ to $t$: $[s_{t-m}, s_{t-m+1}, \ldots, s_t]$. We experimented up to $m = k$: up to $m = 10$ for *DeepMind Lab* and $m = 5$ for *MiniGrid*. The result shows that stacking the observation does not necessarily improves the performance for MDP order greater than 1, which is often observed when the function approximation is used (*e.g.*, Savinov et al. (2018b)). Thus, we did not augment the observation in all the experiments.

# B ANALYSIS ON THE REACHABILITY NETWORK (RNET)

## B.1 ACCURACY OF THE REACHABILITY NETWORK

We measured the accuracy of reachability network on *DeepMind Lab* and *MiniGrid* in Figure 14 and Figure 13. The accuracy was measured on the validation set; we sampled 15,000 positive and negative samples respectively from the replay buffer of size 60,000. Specifically, for an anchor $s_t$, we sample the positive sample $s_{t'}$ from $t' \in [t+1, t+k+\Delta t]$, and the negative sample $s_{t''}$ from $t'' \in [t+k+\Delta t+\Delta_-, t+k+\Delta t+2\Delta_-]$. The RNet reaches the accuracy higher than 80% in

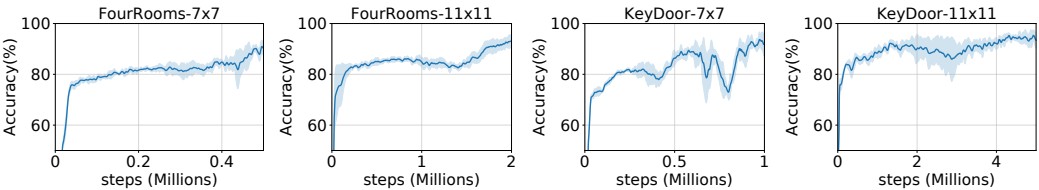

Figure 13: The accuracy of the learned reachability network on (a) *FourRooms-7×7* (b) *FourRooms-11×11*, (c) *KeyDoors-7×7* and (d) *KeyDoors-11×11* in *MiniGrid* in terms of environment steps.

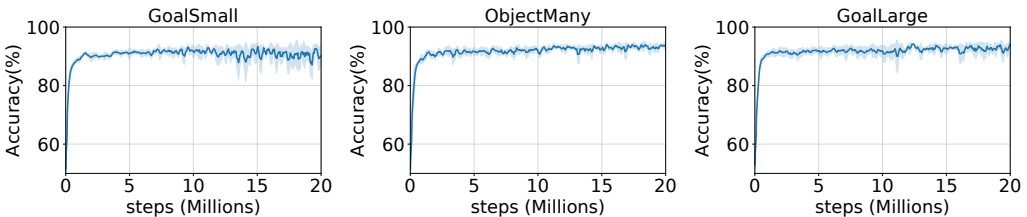

Figure 14: The accuracy of the learned reachability network on (a) *GoalSmall* (b) *ObjectMany*, and (c) *GoalLarge* in *DeepMind Lab* in terms of environment steps.

only 0.4M steps in both *MiniGrid* and *DeepMind Lab*. We note that this is quite high considering the unavoidable noise in the negative samples; *i.e.*, since the negative samples are sampled based on the temporal distance, not based on the actual reachability, they have non-zero probability of being reachable, in which case they are in fact the positive samples.

## B.2 ABLATION STUDY: COMPARISON BETWEEN THE LEARNED RNET AND THE GT-RNET

In this section, we study the effect of RNet's accuracy on the **SPRL**'s performance. To this end, we implement and compare the ground-truth reachability network by computing the ground-truth distance between a pair of states in *MiniGrid*.

**Ground-truth reachability network** was implemented by computing the distance between the two state inputs, and comparing with $k$. For the state inputs $s$ and $s'$, we roll out all possible $k$-step trajectories starting from the state $s$ using the ground-truth single-step forward model. If $s'$ is ever visited during the roll-out, the output of $k$-reachability network is 1 and otherwise, the output is 0.

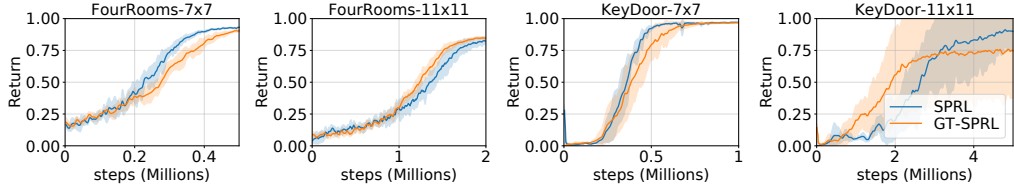

Figure 15: The accuracy of the learned reachability network on (a) *FourRooms-7×7* (b) *FourRooms-11×11*, (c) *KeyDoors-7×7* and (d) *KeyDoors-11×11* in *MiniGrid* in terms of environment steps.

**Result.** We compared the performance of our **SPRL** with the learned RNet and the ground-truth RNet (**GT-SPRL**) in Figure 15 with the best hyperparameters. Overall, the performance of **SPRL**

and **GT-SPRL** are similar. This is partly because the learned RNet achieves a quite high accuracy in early stage of learning (see Figure 13). Interestingly, we can observe that our SPRL with learned RNet performs better than SPRL with GT-RNet on *FourRooms-7×7* and *KeyDoors-7×7*. This is possible since a small noise in RNet output can have a similar effect to the increased tolerance $\Delta t$ on RNet, which makes the resulting cost more dense, which may be helpful depending on the tasks and hyperparameters.

### B.3 ABLATION STUDY: CURRICULUM LEARNING OF RNET

We applied the linear scheduling of $k$ to see if curriculum learning can improve the RNet training, and eventually improve our **SPRL** performance. For example, in case of *FourRooms-7×7*, we used $k = 1$ until 0.1M steps, and $k = 2$ until 0.2M steps, and so on, and finally $k = 5$ from 0.4M steps to 0.5M steps. Figure 16 and Figure 17 compares the performance of **SPRL** on four *MiniGrid* tasks and three *DeepMind Lab* tasks with and without curriculum learning applied respectively. The result shows that, however, the curriculum learning is not helpful for RNet training. We conjecture that changing $k$ during training RNet makes the learning unstable since changing $k$ completely flips the learning target for some inputs; e.g, the two states that are 2 steps away are unreachable for $k = 1$ but reachable for $k = 2$. In Figure 18, we compared the RNet accuracy when RNet was trained with and without curriculum learning. We can observe that curriculum learning achieves similar or lower RNet accuracy with higher standard error, which indicates that the curriculum learning makes RNet training unstable.

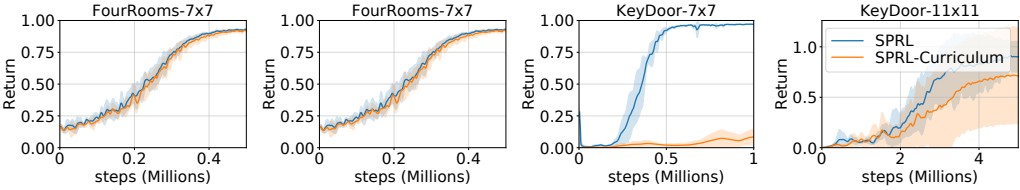

Figure 16: The performance of SPRL with and without curriculum learning of RNet on (a) *FourRooms-7×7* (b) *FourRooms-11×11*, (c) *KeyDoors-7×7* and (d) *KeyDoors-11×11* in *MiniGrid*.

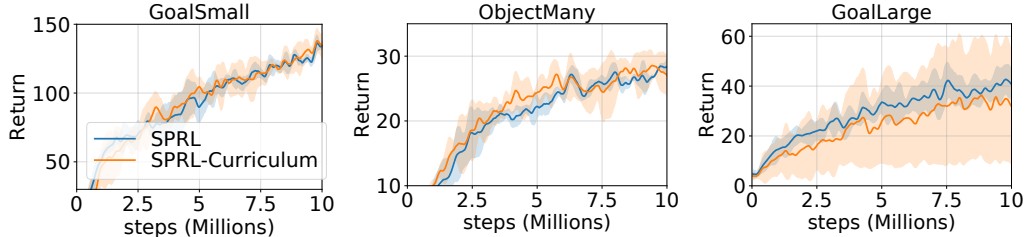

Figure 17: The performance of SPRL with and without curriculum learning of RNet on (a) *GoalSmall* (b) *ObjectMany*, and (c) *GoalLarge* in *DeepMind Lab* in terms of environment steps.

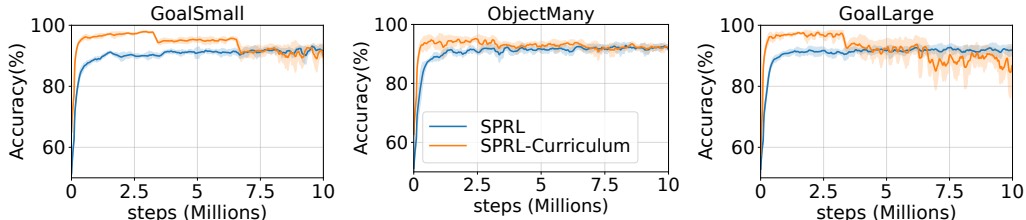

Figure 18: The accuracy of the learned reachability network with and without curriculum learning of RNet on (a) *GoalSmall* (b) *ObjectMany*, and (c) *GoalLarge* in *DeepMind Lab* in terms of environment steps.

## C  EXPERIMENT DETAILS OF *MiniGrid* DOMAIN

### C.1  ENVIRONMENT

*MiniGrid* is a 2D grid-world environment with diverse predefined tasks (Chevalier-Boisvert et al., 2018). It has several challenging features such as pictorial observation, random initialization of the agent and the goal, complex action space and transition dynamics involving agent's orientation of movement and changing object status via interaction (e.g., key-door).

**State Space.**  An observation $\mathbf{s}_t$ is represented as $H \times W \times C$ tensor, where $H$ and $W$ are the height and width of map respectively, and $C$ is features of the objects in the grid. The $(h, w)$-th element of observation tensor is $(type, \ color, \ status)$ of the object and for the coordinate of agent, the $(h, w)$-th element is $(type, \ 0, \ direction)$. The map size (*i.e.*, $H \times W$) varies depending on the task; *e.g.*, for *FourRooms-7×7* task, the map size is $7 \times 7$.

**Action Space and transition dynamics**  The episode terminates in 100 steps, and the episode may terminate earlier if the agent reaches the goal before 100 steps. The action space consists of seven discrete actions with the following transitions.

- `Turn-Counter-Clockwise`: change the $direction$ counter-clockwise by 90 degree.
- `Turn-Clockwise`: change the $direction$ clockwise by 90 degree.
- `Move-Forward`: move toward $direction$ by 1 step unless blocked by other objects.
- `Pick-up-key`: pickup the key if the key is in front of the agent.
- `Drop-the-key`: drop the key in front of the agent.
- `Open/Close-doors`: open/close the door if the door is in front of the agent.
- `Optional-action`: not used

**Reward function.**  The reward is given only if the agent reaches the goal location, and the reward magnitude is $1 - 0.9$(length of episode/maximum step for episode). Thus, the agent can maximize the reward by reaching to the goal location in shortest time.

### C.2  TASKS

In *FourRooms-7×7* and *FourRooms-11×11*, the map structure has four large rooms, and the agent needs to reach to the goal. In *KeyDoors-7×7* and *KeyDoors-11×11*, the agent needs to pick up the key, go to the door, and open the door before reaching to the goal location.

### C.3  ARCHITECTURE AND HYPER-PARAMETERS

We used a simple CNN architecture similar to (Mnih et al., 2015) for policy network. The network consists of `Conv1(16x2x2-1/SAME)-CReLU-Conv2(8x2x2-1/SAME)-CReLU-Conv3(8x2x2-1/SAME)-CReLU-FC(512)-FC(action-dimension)`, where SAME padding ensures the input and output have the same size (*i.e.*, width and height) and CReLU (Shang et al., 2016) is a non-linear activation function applied after each layer. We used Adam (Kingma & Ba, 2014) optimizer to optimize the policy network.

For hyper-parameter search, we swept over a set of hyper-parameters specified in Table 1, and chose the best one in terms of the mean AUC over all the tasks, which is also summarized in Table 1.

## D  EXPERIMENT DETAILS OF *DeepMind Lab* DOMAIN

### D.1  ENVIRONMENT

*DeepMind Lab* is a 3D-game environment with first-person view. Along with random initialization of the agent and the goal, complex action space including directional change, random change of texture, color and maze structure are features that make tasks in *DeepMind Lab* hard to be learned.

**State Space.**  An observation $\mathbf{s}_t$ has the dimension of $120 \times 160 \times 3$ tensor. Observation is given as a first-person view of the map structure.

**Action Space and transition dynamics**  The episode terminates after the fixed number of steps regardless of goal being achieved. The original action space consists of seven discrete actions: `Move-Forward, Move-Backward, Strafe Left, Strafe Right, Look Left, Look Right, Look Left and Move-Forward, Look Right and Move-Forward`. In our experiment, we used eight discrete actions with the additional action `Fire` as in (Higgins et al., 2017; Vezhnevets et al., 2017; Savinov et al., 2018b; Espeholt et al., 2018; Khetarpal & Precup, 2018).

| PPO | | |
|---|---|---|
| **Hyperparameters** | **Sweep range** | **Final value** |
| Learning rate | 0.001, 0.002, 0.003 | 0.003 |
| Entropy | 0.003, 0.005, 0.01, 0.02, 0.05 | 0.01 |
| ICM | | |
| **Hyperparameters** | **Sweep range** | **Final value** |
| Learning rate | 0.001, 0.002, 0.003 | 0.003 |
| Entropy | - | 0.01 |
| Forward/Inverse model loss weight ratio | 0.2, 0.5, 0.8, 1.0 | 0.8 |
| Curiosity module loss weight | 0.03, 0.1, 0.3, 1.0 | 0.3 |
| **ICM** bonus weight | 0.1, 0.3, 1.0, 3.0 | 0.1 |
| GT-Grid | | |
| **Hyperparameters** | **Sweep range** | **Final value** |
| Learning rate | 0.001, 0.002, 0.003 | 0.003 |
| Entropy | - | 0.01 |
| **GT-Grid** bonus weight | 0.003, 0.01, 0.03, 0.1, 0.3 | 0.01 |
| ECO | | |
| **Hyperparameters** | **Sweep range** | **Final value** |
| Learning rate | - | 0.003 |
| Entropy | - | 0.01 |
| $k$ | 3, 5, 7 | 3 |
| **ECO** bonus weight | 0.001, 0.002, 0.005, 0.01 | 0.001 |
| SPRL | | |
| **Hyperparameters** | **Sweep range** | **Final value** |
| Learning rate | 0.003, 0.01 | 0.01 |
| Entropy | - | 0.01 |
| $k$ | 2, 5 | 2 |
| Tolerance ($\Delta t$) | - | 1 |
| Negative bias ($\Delta^-$) | 10, 20 | 20 |
| Positive bias ($\Delta^+$) | - | 5 |
| Cost scale ($\lambda$) | 0.001, 0.002, 0.005 | 0.002 |
| $N_{\Delta t}$ | 30, 60 | 60 |

Table 1: The range of hyperparameters sweeped over and the final hyperparameters used in *MiniGrid* domain.

## D.2 TASKS

We tested our agent and compared methods on three standard tasks in *DeepMind Lab*: *GoalSmall*, *GoalLarge*, and *ObjectMany* which correspond to `explore_goal_locations_small`, `explore_goal_locations_large`, and `explore_object_rewards_many`, respectively. *GoalSmall* and *GoalLarge* has a single goal in the maze, but the size of the maze is larger in *GoalLarge* than *GoalSmall*. The agent and goal locations are randomly set in the beginning of the episode and the episode length is fixed to 1,350 steps for *GoalSmall* and 1,800 steps for *GoalLarge*. When the agent reaches the goal, it positively rewards the agent and the agent is re-spawned in a random location without terminating the episode, such that the agent can reach to the goal multiple times within a single episode. Thus, the agent's goal is to reach to the goal location as many times as possible within the episode length. *ObjectMany* has multiple objects in the maze, where reaching to the object positively rewards the agent and the object disappears. The episode length is fixed to 1,800 steps. The agent's goal is to gather as many object as possible within the episode length.

---

**Algorithm 2** Sampling the triplet data from an episode for RNet training

---

**Require:** Hyperparameters: $k \in \mathbb{N}$, Positive bias $\Delta^+ \in \mathbb{N}$, Negative bias $\Delta^- \in \mathbb{N}$

1: Initialize $t_{\text{anc}} \leftarrow 0$.
2: Initialize $S_{\text{anc}} = \emptyset$, $S_+ = \emptyset$, $S_- = \emptyset$.
3: **while** $t_{\text{anc}} < T$ **do**
4:      $S_{\text{anc}} = S_{\text{anc}} \cup \{s_{t_{\text{anc}}}\}$.
5:      $t_+ = \text{Uniform}(t_{\text{anc}} + 1, t_{\text{anc}} + k)$.
6:      $t_- = \text{Uniform}(t_{\text{anc}} + k + \Delta^-, T)$.
7:      $S_+ = S_+ \cup \{s_{t_+}\}$.
8:      $S_- = S_- \cup \{s_{t_-}\}$.
9:      $t_{\text{anc}} = \text{Uniform}(t_+ + 1, t_+ + \Delta^+)$.
    **return** $S_{\text{anc}}, S_+, S_-$

---

### D.3   REACHABILITY NETWORK TRAINING

Similar to Savinov et al. (2018b), we used the following contrastive loss for training the reachability network:

$$\mathcal{L}_{\text{Rnet}} = -\log\left(\text{Rnet}_{k-1}(s_{\text{anc}}, s_+)\right) - \log\left(1 - \text{Rnet}_{k-1}(s_{\text{anc}}, s_-)\right), \tag{15}$$

where $s_{\text{anc}}, s_+, s_-$ are the anchor, positive, and negative samples, respectively. The anchor, positive and negative samples are sampled from the same episode, and their time steps are sampled according to Algorithm 2. The RNet is trained in an off-policy manner from the replay buffer with the size of 60K environment steps collecting agent's online experience. We found that adaptive scheduling of RNet is helpful for faster convergence of RNet. Out of 20M total environment steps, for the first 1M, 1M, and 18M environment steps, we updated RNet every $6K$, $12K$, and $36K$ environment steps, respectively. For all three environments of *DeepMind Lab*, RNet accuracy was $\sim 0.9$ after 1M steps.

**Multiple tolerance.** In order to improve the stability of Reachability prediction, we used the statistics over multiple samples rather than using a single-sample estimate as suggested in Eq. (13). As a choice of sampling method, we simply used multiples of tolerance. In other words, given $s_{t-(k+\Delta t)}$ and $s_t$ as inputs for reachability network, we instead used $s_{t-(k+n\Delta t)}$ and $s_t$ where $1 \leq n \leq N_{\Delta t}$, $n \in \mathbb{N}$ and $N_{\Delta t}$ is the number of tolerance samples. We used 90-percentile of $N_{\Delta t}$ outputs of reachability network, $\text{Rnet}_{k-1}(s_{t-(k+n\Delta t)}, s_t)$, as in (Savinov et al., 2018b) to get the representative of the samples.

### D.4   ARCHITECTURE AND HYPER-PARAMETERS

Following (Savinov et al., 2018b), we used the same CNN architecture used in (Mnih et al., 2015). To fit the architecture, we resized the input observation image into $84 \times 84 \times 3$ image, and normalized by dividing the pixel value by 255.

For **SPRL**, we used a smaller reachability network (RNet) architecture compared to **ECO** to reduce the training time. The RNet is based on siamese architecture with two branches. Following (Savinov et al., 2018b), **ECO** used Resnet-18 (He et al., 2016) architecture with 2-2-2-2 residual blocks and 512-dimensional output fully-connected layer to implement each branch. For **SPRL**, we used Resnet-12 with 2-2-1 residual blocks and 512-dimensional output fully-connected layer to implement each branch. The RNet takes two states as inputs, and each state is fed into each branch. The outputs of the two branches are concatenated and forwarded to three[4] 512-dimensional fully-connected layers to produce one-dimensional sigmoid output, which predicts the reachability between two state inputs. We also resized the observation to the same dimension as policy (*i.e.*, $84 \times 84 \times 3$, which is smaller than the original $120 \times 160 \times 3$ used in (Savinov et al., 2018b)).

For all the baselines (*i.e.*, **PPO**, **ECO**, **ICM**, and **GT-Grid**), we used the best hyperparameter used in (Savinov et al., 2018b). For **SPRL**, we searched over a set of hyperparameters specified in Table 2, and chose the best one in terms of the mean AUC over all the tasks, which is also summarized in Table 2.

---

[4]Savinov et al. (2018b) used four 512-dimensional fully-connected layers.

| Hyperparameters for SPRL | Sweep range | Final value |
| --- | --- | --- |
| Learning rate | - | 0.0003 |
| Entropy | - | 0.004 |
| $k$ | 3, 10, 30 | 10 |
| Tolerance ($\Delta t$) | 1, 3, 5 | 1 |
| Negative bias ($\Delta^-$) | 5, 10, 20 | 20 |
| Positive bias ($\Delta^+$) | - | 5 |
| Cost scale ($\lambda$) | 0.02, 0.06, 0.2 | 0.06 |
| Optimizer | - | Adam |
| $N_{\Delta t}$ | - | 200 |

Table 2: The range of hyperparameters sweeped over and the final hyperparameters used for our **SPRL** method in *DeepMind Lab* domain.

# E  OPTION FRAMEWORK-BASED FORMULATION

## E.1  PRELIMINARY: OPTION FRAMEWORK

Options framework (Sutton, 1998) defines options as a generalization of actions to include temporally extended series of action. Formally, options consist of three components: a policy $\pi : \mathcal{S} \times \mathcal{A} \to [0, 1]$, a termination condition $\beta : \mathcal{S}^+ \to [0, 1]$, and an initiation set $\mathcal{I} \subseteq \mathcal{S}$. An option $\langle \mathcal{I}, \pi, \beta \rangle$ is available in state $s$ if and only if $s \in \mathcal{I}$. If the option is taken, then actions are selected according to $\pi$ until the option terminates stochastically according to $\beta$. Then, the option-reward and option-transition models are defined as

$$r_s^o = \mathbb{E} \left\{ r_{t+1} + \gamma r_{t+2} + \cdots + \gamma^{k-1} r_{t+k} \mid E(o, s, t) \right\} \tag{16}$$

$$P_{ss'}^o = \sum_{k=1}^{\infty} p \left( s', k \right) \gamma^k \tag{17}$$

where $t + k$ is the random time at which option $o$ terminates, $E(o, s, t)$ is the event that option $o$ is initiated in state $s$ at time $t$, and $p(s', k)$ is the probability that the option terminates in $s'$ after $k$ steps. Using the option models, we can re-write Bellman equation as follows:

$$V^\pi(s) = \mathbb{E} \left[ r_{t+1} + \cdots + \gamma^{k-1} r_{t+k} + \gamma^k V^\pi(s_{t+k}) \right], \tag{18}$$

$$= \sum_{o \in \mathcal{O}} Pr[E(o, s)] \left[ r_s^o + \sum_{s'} P_{ss'}^o V^\pi(s') \right]. \tag{19}$$

where $t + k$ is the random time at which option $o$ terminates and $E(o, s)$ is the event that option $o$ is initiated in state $s$.

## E.2  OPTION-BASED VIEW-POINT OF SHORTEST-PATH CONSTRAINT

In this section, we present an option framework-based viewpoint of our shortest-path (SP) constraint. We will first show that a (sparse-reward) MDP can be represented as a weighted directed graph where nodes are rewarding states, and edges are options. Then, we show that a policy satisfying SP constraint also maximizes the option-transition probability $P_{ss'}^o$.

For a given MDP $\mathcal{M} = (\mathcal{S}, \mathcal{A}, \mathcal{R}, \mathcal{P}, \rho, \bar{\mathcal{S}})$, let $\mathcal{S}^R = \{s | R(s) \neq 0\} \subset \mathcal{S}$ be the set of all rewarding states, where $R(s)$ is the reward function upon arrival to state $s$. In sparse-reward tasks, it is assumed that $|\mathcal{S}^R| << |\mathcal{S}|$. Then, we can form a weighted directed graph $G^\pi = (\mathcal{V}, \mathcal{E})$ of policy $\pi$ and given MDP. The vertex set is defined as $\mathcal{V} = \mathcal{S}^R \cup \rho_0 \cup \bar{\mathcal{S}}$ where $\mathcal{S}^R$ is rewarding states, $\rho_0$ is the initial states, and $\bar{\mathcal{S}}$ is the terminal states. Similar to the path set in Definition 2, let $\mathcal{T}_{s \to s'}$ denotes a set of paths transitioning from one vertex $s \in \mathcal{V}$ to another vertex $s' \in \mathcal{V}$:

$$\mathcal{T}_{s \to s'} = \{\tau | s_0 = s, s_{\ell(\tau)} = s', \{s_t\}_{0 < t < \ell(\tau)} \cap \mathcal{V} = \emptyset\}. \tag{20}$$

Then, the edge from a vertex $s \in \mathcal{V}$ to another vertex $s' \in \mathcal{V}$ is defined by an (implicit) option tuple: $o(s, s') = (\mathcal{I}, \pi, \beta)_{(s, s')}$, where $\mathcal{I} = \{s\}$, $\beta(s) = \mathbb{I}(s = s')$, and

$$\pi^{(s, s')}(\tau) = \begin{cases} \frac{1}{Z} \pi(\tau) & \text{for } \tau \in \mathcal{T}_{s \to s'} \\ 0 & \text{otherwise} \end{cases}, \tag{21}$$

where $Z$ is the partition function to ensure $\int \pi^{(s, s')}(\tau) d\tau = 1$. Following Eq. (16), the option-reward is given as

$$r_{s, s'}^\pi = \mathbb{E}^{\pi^{(s, s')}} \left[ r_{t+1} + \gamma r_{t+2} + \cdots + \gamma^{k-1} r_{t+k} \mid E(o^{(s, s')}, s, t) \right], \tag{22}$$

$$= \mathbb{E}^{\pi^{(s, s')}} \left[ \gamma^{k-1} r_{t+k} \mid E(o^{(s, s')}, s, t) \right], \tag{23}$$

where $t + k$ is the random time at which option $o(s, s')$ terminates, and $E(o, s, t)$ is the event that option $o(s, s')$ is initiated in state $s$ at time $t$. Note that in the last equality, $r_{t+1} = \cdots = r_{t+k-1} = 0$ holds since $\{s_{t+1}, \ldots, s_{t+k-1}\} \cap \mathcal{V} = \emptyset$ from the definition of option policy $\pi^{(s, s')}$. Following

Eq. (17), the option transition is given as

$$P_{s,s'}^{\pi} = \sum_{k=1}^{\infty} p(s', k)\gamma^k \tag{24}$$

$$= \mathbb{E}^{\pi}\left[\gamma^k | s_0 = s, s_k = s', \{r_t\}_{t<k} = 0\right] \tag{25}$$

$$= \gamma^{D_{\mathrm{nr}}^{\pi}(s,s')}. \tag{26}$$

where $p(s', k)$ is the probability that the option terminates in $s'$ after $k$ steps, and $D_{\mathrm{nr}}^{\pi}(s, s')$ is the $\pi$-distance in Definition 3. Then, we can re-write the shortest-path constraint in terms of $P_{s,s'}^{\pi}$ as follows:

$$\Pi^{\mathrm{SP}} = \{\pi | \forall (s, s' \in \mathcal{T}_{\hat{s}, \hat{s}', \mathrm{nr}}^{\pi} \text{ s.t. } (\hat{s}, \hat{s}') \in \Phi^{\pi}),\ D_{\mathrm{nr}}^{\pi}(s, s') = \min_{\pi} D_{\mathrm{nr}}^{\pi}(s, s')\} \tag{27}$$

$$= \{\pi | \forall (s, s' \in \mathcal{T}_{\hat{s}, \hat{s}', \mathrm{nr}}^{\pi} \text{ s.t. } (\hat{s}, \hat{s}') \in \Phi^{\pi}),\ P_{s,s'}^{\pi} = \max_{\pi} P_{s,s'}^{\pi}\} \tag{28}$$

Thus, we can see that the policy satisfying SP constraint also maximizes the option-transition probability. We will use this result in Appendix F.

# F   SHORTEST-PATH CONSTRAINT: A SINGLE-GOAL CASE

In this section, we provide more discussion on a special case of the shortest-path constraint (Section 3.1), when the (stochastic) MDP defines a single-goal task: *i.e.*, there exists a unique initial state $s_{\mathrm{init}} \in \mathcal{S}$ and a unique goal state $s_g \in \mathcal{S}$ such that $s_g$ is a terminal state, and $R(s) > 0$ if and only if $s = s_g$.

We first note that the non-rewarding path set is identical to the path set in such a setting, because the condition $r_t = 0 (t < \ell(\tau))$ from Definition 2 is always satisfied as $R(s) > 0 \Leftrightarrow s = s_g$ and $s_{\ell(\tau)} = s_g$:

$$\mathcal{T}_{s,s',\mathrm{nr}}^{\pi} = \mathcal{T}_{s,s'}^{\pi} = \{\tau \mid s_0 = s, s_{\ell(\tau)} = s', p_{\pi}(\tau) > 0, \{s_t\}_{t<\ell(\tau)} \neq s'\} \tag{29}$$

Again, $\mathcal{T}_{s,s'}^{\pi}$ is a set of all path starting from $s$ (*i.e.*, and ending at $s'$ (*i.e.*, $s_{\ell(\tau)} = s'$) where the agent visits $s'$ *only* at the end (*i.e.*, $\{s_t\}_{t<\ell(\tau)} \neq s'$), that can be rolled out by policy with a non-zero probability (*i.e.*, $p_{\pi}(\tau) > 0$).

We now claim that an optimal policy satisfies the shortest-path constraint. The idea is that, since $s_g$ is the only rewarding and terminal state, maximizing $R(\tau) = \gamma^T R(s_g)$ where $s_T = s_g$ corresponds to minimizing the number of time steps $T$ to reach $s_g$. In this setting, a shortest-path policy is indeed optimal.

**Lemma 4.** *For a single-goal MDP, any optimal policy satisfies the shortest-path constraint.*

*Proof.* Let $s_{\mathrm{init}}$ be the initial state and $s_g$ be the goal state. We will prove that any optimal policy is a shortest-path policy from the initial state to the goal state. We use the fact that $s_g$ is the only rewarding state, *i.e.*, $R(s) > 0$ entails $s = s_g$.

$$\pi^* = \arg\max_{\pi} \mathbb{E}_{s\sim\rho}^{\tau\sim\pi}\left[\sum_t \gamma^t r_t \,\bigg|\, s_0 = s\right] \tag{30}$$

$$= \arg\max_{\pi} \mathbb{E}^{\tau\sim\pi}\left[\sum_t \gamma^t r_t \,\bigg|\, s_0 = s_{\mathrm{init}}\right] \tag{31}$$

$$= \arg\max_{\pi} \mathbb{E}^{\tau\sim\pi}\left[\gamma^T R(s_g) \mid s_0 = s_{\mathrm{init}}, s_{\ell(\tau)} = s_g\right] \tag{32}$$

$$= \arg\max_{\pi} \mathbb{E}^{\tau\sim\pi}\left[\gamma^T \mid s_0 = s_{\mathrm{init}}, s_{\ell(\tau)} = s_g\right] \tag{33}$$

$$= \arg\min_{\pi} \log_{\gamma}\left(\mathbb{E}^{\tau\sim\pi}\left[\gamma^T \mid s_0 = s_{\mathrm{init}}, s_{\ell(\tau)} = s_g\right]\right) \tag{34}$$

$$= \arg\min_{\pi} D_{\mathrm{nr}}^{\pi}(s_{\mathrm{init}}, s_g), \tag{35}$$

where Eq. (33) holds since $R(s_g) > 0$ from our assumption that $R(s) + V^*(s) > 0$. $\qquad\square$

# G    PROOF OF THEOREM 1

We make the following assumptions on the Markov Decision Process (MDP) $\mathcal{M}$: namely *mild stochasticity* (Definitions 8 and 9).

**Definition 8** (Mild stochasticity (1)). *In MDP $\mathcal{M}$, there exists an optimal policy $\pi^*$ and the corresponding shortest-path policy $\pi^{sp} \in \Pi^{SP}$ such that for all $s, s' \in \Phi^\pi$, it holds $p_{\pi^*}(\bar{s} = s'|s_0 = s) = p_{\pi^{sp}}(\bar{s} = s'|s_0 = s)$.*

**Definition 9** (Mild stochasticity (2)). *In MDP $\mathcal{M}$, the optimal policy $\pi^*$ does not visit the same state more than once: For all $s \in \mathcal{S}$ such that $\rho_{\pi^*}(s) > 0$, it holds $\rho_{\pi^*}(s) = 1$, where $\rho_\pi(s) \triangleq \mathbb{E}_{s_0 \sim \rho_0(S), a \sim \pi(A|s), s' \sim (S|s,a)} \left[ \sum_{t=1}^T \mathbb{I}(s_t = s) \right]$ is the state-visitation count.*

In other words, we assume that the optimal policy does not have a cycle. One common property of MDP that meets this condition is that the reward disappearing after being acquired by the agent. We note that this assumption holds for many practical environments. In fact, in many cases as well as *Atari*, *DeepMind Lab*, etc.

**Theorem 1.** *For any MDP with the mild stochasticity condition, an optimal policy $\pi^*$ satisfies the shortest-path constraint: $\pi^* \in \Pi^{SP}$.*

*Proof.* For simplicity, we prove this based on the option-based view point (see Appendix E). By plugging Eq. (23) and Eq. (25) into Eq. (19), we can re-write the Bellman equation of the value function $V^\pi(s)$ as follows:

$$V^\pi(s) = \sum_{o \in \mathcal{O}} Pr[E(o,s)] \left[ r_s^o + \sum_{s'} P_{ss'}^o V^\pi(s') \right] \tag{36}$$

$$= \sum_{s' \in \mathcal{S}^{\text{IR}}} p_\pi(\bar{s} = s'|s_0 = s) \left[ R(s') \mathbb{E}^{\tau \sim \pi}(\gamma^{\ell(\tau)}|s_0 = s, \bar{s} = s') + \gamma P_{s,s'}^\pi V^\pi(s') \right] \tag{37}$$

$$= \sum_{s' \in \mathcal{S}^{\text{IR}}} p_\pi(\bar{s} = s'|s_0 = s) \left[ R(s') P_{s,s'}^\pi + \gamma P_{s,s'}^\pi V^\pi(s') \right], \tag{38}$$

$$= \sum_{s' \in \mathcal{S}^{\text{IR}}} p_\pi(\bar{s} = s'|s_0 = s) P_{s,s'}^\pi \left[ R(s') + \gamma V^\pi(s') \right], \tag{39}$$

where $\bar{s}$ is the first rewarding state that agent encounters. Intuitively, $p_\pi(\bar{s} = s'|s_0 = s)$ means the probability that the $s'$ is the first rewarding state that policy $\pi$ encounters when it starts from $s$. From Eq. (28), our goal is to show:

$$\pi^* \in \Pi^{SP} = \{\pi \mid \forall(s, s') \in \mathcal{T}_{\Phi,\text{nr}}^\pi, P_{s,s'}^\pi = P_{s,s'}^* \}, \tag{40}$$

where $P_{s,s'}^* = \max_\pi P_{s,s'}^\pi$.

We will prove Eq. (40) by contradiction. Suppose $\pi^*$ is an optimal policy such that $\pi^* \notin \Pi^{SP}$. Then,

$$\exists(\hat{s}, \hat{s}' \in \mathcal{T}_{\Phi,\text{nr}}^{\pi^*}) \text{ s.t. } P_{\hat{s},\hat{s}'}^{\pi^*} \neq P_{\hat{s},\hat{s}'}^*. \tag{41}$$

Recall the definition: $P_{s,s'}^* = \max_\pi P_{s,s'}^\pi$. Then, for any $\pi$, the following statement is true.

$$P_{s,s'}^\pi \neq P_{s,s'}^* \leftrightarrow P_{s,s'}^\pi < P_{s,s'}^*. \tag{42}$$

Thus, we have

$$P_{\hat{s},\hat{s}'}^{\pi^*} < P_{\hat{s},\hat{s}'}^* \tag{43}$$

Let $\pi_{sp} \in \Pi^{SP}$ be a shortest path policy that preserves stochastic dynamics from Definition 8. Then, we have

$$P_{\hat{s},\hat{s}'}^{\pi^*} < P_{\hat{s},\hat{s}'}^* = P_{\hat{s},\hat{s}'}^{\pi_{sp}}. \tag{44}$$

Then, let's compose a new policy $\hat{\pi}$:

$$\hat{\pi}(a|s) = \begin{cases} \pi_{sp}(a|s) & \text{if } \exists \tau \in \mathcal{T}_{\hat{s},\hat{s}',\text{nr}}^{\pi_{sp}} \text{ s.t. } s \in \tau \\ \pi^*(a|s) & \text{otherwise} \end{cases}. \tag{45}$$

Now consider a path $\tau_{\hat{s} \to \hat{s}'}$ that agent visits $\hat{s}$ at time $t = i$ and transitions to $\hat{s}'$ at time $t = j > i$ while not visiting any rewarding state from $t = i$ to $t = j$ with non-zero probability (*i.e.*, $p_{\pi_{sp}}(\tau) > 0$).

We can define a set of such paths as follows:

$$\hat{\mathcal{T}}_{\hat{s}\to\hat{s}'} = \{\tau \mid \exists (i < j), s_i = \hat{s}, s_j = \hat{s}', \{s_t\}_{i<t<j} \cap \mathcal{S}^{\text{IR}} = \emptyset, p_{\pi_{\text{sp}}}(\tau) > 0\}. \tag{46}$$

To reiterate the definitions from Definition 6: $\mathcal{S}^{\text{IR}} = \{s \mid R(s) > 0 \text{ or } \rho(s) > 0\}$ is the union of all initial and rewarding states, and $\Phi^\pi = \{(s, s') \mid s, s' \in \mathcal{S}^{\text{IR}}, \rho(s) > 0, \mathcal{T}^\pi_{s,s',\text{nr}} \neq \emptyset\}$ is the subset of $\mathcal{S}^{\text{IR}}$ such that agent may roll out.

From Definition 9 and Eq. (45), the likelihood of a path $\tau$ under policy $\hat{\pi}$ is given as follows:

$$p_{\hat{\pi}}(\tau) = \begin{cases} p_{\pi^*}(\tau \in \hat{\mathcal{T}}_{\hat{s}\to\hat{s}'}) p_{\pi_{\text{sp}}}(\tau \mid \tau \in \hat{\mathcal{T}}_{\hat{s}\to\hat{s}'}) & \text{for } \tau \in \hat{\mathcal{T}}_{\hat{s}\to\hat{s}'} \\ p_{\pi^*}(\tau) & \text{otherwise} \end{cases}, \tag{47}$$

where $p_{\hat{\pi}}(\tau)$ is the likelihood of trajectory $\tau$ under policy $\hat{\pi}$, $p_{\hat{\pi}}(\tau \in \hat{\mathcal{T}}_{\hat{s}\to\hat{s}'}) = \int_{\tau\in\hat{\mathcal{T}}_{\hat{s}\to\hat{s}'}} p_{\hat{\pi}}(\tau)d\tau$ ensures the likelihood $\hat{\pi}(\tau)$ to be a valid probability density function (*i.e.*, $\int p_{\hat{\pi}}(\tau)d\tau = 1$). From the path $\tau_{\hat{s}\to\hat{s}'}$ and $i, j$, we will choose two states $s_{\text{ir}}, s'_{\text{ir}} \sim \tau_{\hat{s}\to\hat{s}'}$, where

$$s_{\text{ir}} = \max_t (s_t \mid s_t \in \mathcal{S}^{\text{IR}}, t \leq i), \quad s'_{\text{ir}} = \min_t (s_t \mid s_t \in \mathcal{S}^{\text{IR}}, j \leq t). \tag{48}$$

Note that such $s_{\text{ir}}$ and $s'_{\text{ir}}$ always exist in $\tau_{\hat{s}\to\hat{s}'}$ since the initial state and the terminal state satisfy the condition to be $s_{\text{ir}}$ and $s'_{\text{ir}}$.

Then, we can show that the path between $s_{\text{ir}}$ and $s'_{\text{ir}}$ is **not** a shortest-path. Recall the definition of $D^\pi_{\text{nr}}(s, s')$ (Definition 3):

$$D^{\pi^*}_{\text{nr}}(s_{\text{ir}}, s'_{\text{ir}}) := \log_\gamma \left( \mathbb{E}_{\tau\sim\pi^*: \tau\in\mathcal{T}^{\pi^*}_{s_{\text{ir}},s'_{\text{ir}},\text{nr}}} \left[ \gamma^{\ell(\tau)} \right] \right) \tag{49}$$

$$= \log_\gamma \left( \mathbb{E}_{\tau\sim\pi^*} \left[ \underbrace{\gamma^{\ell(\tau)} \mid \tau \in \mathcal{T}^{\pi^*}_{s_{\text{ir}},s'_{\text{ir}},\text{nr}}}_{\clubsuit} \right] \right) \tag{50}$$

where we will use $\clubsuit := \gamma^{\ell(\tau)} \mid \tau \in \mathcal{T}^{\pi^*}_{s_{\text{ir}},s'_{\text{ir}},\text{nr}}$ for a shorthand notation. Then, we have

$$\gamma^{D^{\pi^*}_{\text{nr}}(s_{\text{ir}},s'_{\text{ir}})} := \mathbb{E}_{\tau\sim\pi^*}[\clubsuit] \tag{51}$$

$$= p_{\pi^*}(\tau \in \hat{\mathcal{T}}_{\hat{s}\to\hat{s}'}) \mathbb{E}_{\tau\sim\pi^*} \left[ \clubsuit \mid \tau \in \hat{\mathcal{T}}_{\hat{s}\to\hat{s}'} \right]$$

$$+ p_{\pi^*}(\tau \notin \hat{\mathcal{T}}_{\hat{s}\to\hat{s}'}) \mathbb{E}_{\tau\sim\pi^*} \left[ \clubsuit \mid \tau \notin \hat{\mathcal{T}}_{\hat{s}\to\hat{s}'} \right] \tag{52}$$

$$\text{(From Definition 5)} \quad < p_{\pi^*}(\tau \in \hat{\mathcal{T}}_{\hat{s}\to\hat{s}'}) \mathbb{E}_{\tau\sim\pi_{\text{sp}}} \left[ \clubsuit \mid \tau \in \hat{\mathcal{T}}_{\hat{s}\to\hat{s}'} \right]$$

$$+ p_{\pi^*}(\tau \notin \hat{\mathcal{T}}_{\hat{s}\to\hat{s}'}) \mathbb{E}_{\tau\sim\pi^*} \left[ \clubsuit \mid \tau \notin \hat{\mathcal{T}}_{\hat{s}\to\hat{s}'} \right] \tag{53}$$

$$\text{(From Eq. (47))} \quad = p_{\hat{\pi}}(\tau \in \hat{\mathcal{T}}_{\hat{s}\to\hat{s}'}) \mathbb{E}_{\tau\sim\hat{\pi}} \left[ \clubsuit \mid \tau \in \hat{\mathcal{T}}_{\hat{s}\to\hat{s}'} \right]$$

$$+ p_{\hat{\pi}}(\tau \notin \hat{\mathcal{T}}_{\hat{s}\to\hat{s}'}) \mathbb{E}_{\tau\sim\hat{\pi}} \left[ \clubsuit \mid \tau \notin \hat{\mathcal{T}}_{\hat{s}\to\hat{s}'} \right] \tag{54}$$

$$= \mathbb{E}_{\tau\sim\hat{\pi}}[\clubsuit] = \gamma^{D^{\hat{\pi}}_{\text{nr}}(s_{\text{ir}},s'_{\text{ir}})} \tag{55}$$

$$\iff D^{\pi^*}_{\text{nr}}(s_{\text{ir}}, s'_{\text{ir}}) > D^{\hat{\pi}}_{\text{nr}}(s_{\text{ir}}, s'_{\text{ir}}) \tag{56}$$

where Ineq. (56) is given by the fact that $\gamma < 1$. Then, $P^{\pi^*}_{s_{\text{ir}},s'_{\text{ir}}} < P^{\hat{\pi}}_{s_{\text{ir}},s'_{\text{ir}}}$.

From Eq. (39), we have

$$V^{\hat{\pi}}(s_{\mathrm{ir}}) = \sum_{s' \in \mathcal{S}^{\mathrm{IR}}} p_{\hat{\pi}}(\bar{s} = s' \mid s_0 = s_{\mathrm{ir}}) P^{\hat{\pi}}_{s_{\mathrm{ir}}, s'} \left[ R(s') + \gamma V^{\hat{\pi}}(s') \right] \tag{57}$$

$$= p_{\hat{\pi}}(\bar{s} = s'_{\mathrm{ir}} \mid s_0 = s_{\mathrm{ir}}) P^{\hat{\pi}}_{s_{\mathrm{ir}}, s'_{\mathrm{ir}}} \left[ R(s'_{\mathrm{ir}}) + \gamma V^{\hat{\pi}}(s'_{\mathrm{ir}}) \right]$$
$$+ \sum_{s' \in \mathcal{S}^{\mathrm{IR}} \setminus s'_{\mathrm{ir}}} p_{\hat{\pi}}(\bar{s} = s' \mid s_0 = s_{\mathrm{ir}}) P^{\hat{\pi}}_{s_{\mathrm{ir}}, s'} \left[ R(s') + \gamma V^{\hat{\pi}}(s') \right] \tag{58}$$

$$= p_{\pi^*}(\bar{s} = s'_{\mathrm{ir}} \mid s_0 = s_{\mathrm{ir}}) P^{\hat{\pi}}_{s_{\mathrm{ir}}, s'_{\mathrm{ir}}} \left[ R(s'_{\mathrm{ir}}) + \gamma V^{\pi^*}(s'_{\mathrm{ir}}) \right]$$
$$+ \sum_{s' \in \mathcal{S}^{\mathrm{IR}} \setminus s'_{\mathrm{ir}}} p_{\pi^*}(\bar{s} = s' \mid s_0 = s_{\mathrm{ir}}) P^{\pi^*}_{s_{\mathrm{ir}}, s'} \left[ R(s') + \gamma V^{\pi^*}(s') \right] \tag{59}$$

$$> p_{\pi^*}(\bar{s} = s'_{\mathrm{ir}} \mid s_0 = s_{\mathrm{ir}}) P^{\pi^*}_{s_{\mathrm{ir}}, s'_{\mathrm{ir}}} \left[ R(s'_{\mathrm{ir}}) + \gamma V^{\pi^*}(s'_{\mathrm{ir}}) \right]$$
$$+ \sum_{s' \in \mathcal{S}^{\mathrm{IR}} \setminus s'_{\mathrm{ir}}} p_{\pi^*}(\bar{s} = s' \mid s_0 = s_{\mathrm{ir}}) P^{\pi^*}_{s_{\mathrm{ir}}, s'} \left[ R(s') + \gamma V^{\pi^*}(s') \right] \tag{60}$$

$$= \sum_{s' \in \mathcal{S}^{\mathrm{IR}}} p_{\pi^*}(\bar{s} = s' \mid s_0 = s_{\mathrm{ir}}) P^{\pi^*}_{s_{\mathrm{ir}}, s'} \left[ R(s') + \gamma V^{\pi^*}(s') \right] \tag{61}$$

$$= V^*(s_{\mathrm{ir}}), \tag{62}$$

where Eq. (59) holds from the *mild-stochasticity (1)* and *mild-stochasticity (2)* assumption, and Ineq. (60) holds because $P^{\hat{\pi}}_{s_{\mathrm{ir}}, s'_{\mathrm{ir}}} > P^{\pi^*}_{s_{\mathrm{ir}}, s'_{\mathrm{ir}}}$ and $R(s') + \gamma V^{\pi^*}(s') > 0$ from the non-negative optimal value assumption (See Section 2). Finally, this is a contradiction since the optimal value function $V^*(s)$ should be the maximum. $\qquad\square$

## H EXTENDED RELATED WORKS

**Approximate state abstraction.** The approximate state abstraction approaches investigate partitioning an MDP's state space into clusters of similar states while preserving the optimal solution. Researchers have proposed several state similarity metrics for MDPs. Dean et al. (2013) proposed to use the bisimulation metrics (Givan et al., 2003; Ferns et al., 2004), which measures the difference in transition and reward function. Bertsekas et al. (1988) used the magnitude of Bellman residual as a metric. Abel et al. (2016; 2018); Li et al. (2006) used the different types of distance in optimal Q-value to measure the similarity between states to bound the sub-optimality in optimal value after the abstraction. Recently, Castro (2019) extended the bisimulation metrics to the approximate version for deep-RL setting where tabular representation of state is not available.

Our shortest-path constraint can be seen as a form of state abstraction, in that ours also aim to reduce the size of MDP (*i.e.*, state and action space) while preserving the "solution quality". However, our method does so by removing sub-optimal policies, not by aggregating similar states (or policies).

**Connection to Option framework** Our shortest-path constraint constrains the policy space to a set of shortest-path policies (See Definition 5 for definition) between initial and rewarding states. It can be seen as a set of options (Sutton, 1998) transitioning between initial and rewarding states. We refer the readers to Appendix E for the detailed description about option framework-based formulation of our framework.

