# OpenReview forum: "Shortest-Path Constrained Reinforcement Learning for Sparse Reward Tasks"
_ICLR.cc/2021/Conference — Reject_

### Official Review · AnonReviewer2 · 2020-10-27
**Clean and intuitive method preventing over-exploration for sparse reward setting**

**Rating:** 6
**Confidence:** 3

**Review:**

The paper proposes a novel k-shortest-path constraint that prevents over-exploration by exploit the combinatorial structure (i.e. shortest path) of sparse reward tasks. It is proved theoretically that the k-shortest-path constraint maintains the optimal policy. Empirical evidence shows that the k-SP constraint indeed significantly improves the sample complexity over baseline algorithms in benchmarking environments.

While I certainly agree that sparse reward tasks is important, one of the biggest challenges in sparse reward tasks is when the environment has a large state space. When the state space is finite and small, there is no doubt that many algorithms (even  algorithms with no deep learning method) can solve the task. However, for a large state space and sparse reward task (such as the robotics tasks in Mujoco environments[1], Montezuma's Revenge in Atari environments), I'm not sure whether the k-SP constraint is enough for efficiently solve the task. The k-SP constraint essentially tells the agent to avoid going to duplicated states. Even so, due to large state space, it is unlikely that the agent can observe non-zero reward without extra guidance. To be more specific, could the author(s) provide more discussion compared to HER (Hindsight Experience Replay) and hierarchical reinforcement learning?

Overall, the paper is well-written, and the empirical results are clear. Therefore, I would recommend a weak acceptance.

[1] http://gym.openai.com/envs/#robotics

---

> ### Author Response · Authors · 2020-11-18
> **Clarification on whether SPRL's goal is to avoid duplicated states**
>
> We appreciate AnonReviewer2 for providing constructive feedback. Please refer to the common response above as well as the individual responses below.
>
> **Q**: “k-SP constraint is essentially about avoiding duplicate states.”
>
> **A**: We note that the goal of k-SP constraint is **fundamentally different** from avoiding the duplicate states which is the main goal of novelty-seeking exploration methods (ICM, RND [Burda et al., 2019], ECO, etc). Our k-SP constraint avoids visiting any state that is not on the k-shortest-path between rewarding states **even though the state has never been visited before** (i.e., novel state).
> We can also see the empirical difference in the trajectories of SPRL and GT-UCB in Figure 6. SPRL makes the agent take the shortest-path trajectory between rewarding states while avoiding other detouring (not duplicated) states. On the other hand, the GT-UCB, which aims to avoid visiting duplicated states, visits every state uniformly and it results in a different trajectory from SPRL.
>
> [1] Burda et al. "Exploration by random network distillation." ICLR. 2019.

---

> > ### Comment · AnonReviewer2 · 2020-11-24
> > **Thanks for addressing my concerns!**
> >
> > Thanks to the author(s) for addressing my concerns. After reading the rebuttal my evaluation remains the same, and I'd like to recommend a weak acceptance.

---

### Official Review · AnonReviewer4 · 2020-10-29
**Interesting work**

**Rating:** 6
**Confidence:** 5

**Review:**

This paper proposes the k-Shortest-Path (k-SP) constraint to restrict the agent’s trajectory to avoid redundant exploration and thus improves sample efficiency in sparse-reward MDPs. Specifically, k-SP constraint is applied to a trajectory rolled out by a policy where all of its sub-path of length k is required to be a shortest-path under the π-distance metric. Instead of a hard constraint, a cost function-based formulation is proposed to implement the constraint. The method can improve the sample efficiency in sparse reward tasks and also preserve the optimality of given MDP. Numerical results in the paper also demonstrate the effectiveness of k-SP compared with existing methods on two domains (1) Mini-Grid and (2) DeepMind Lab in sparse reward settings.

Overall, the paper is well written and clearly conveys the main idea and the main results of the work. The idea and motivation of the paper are very intuitive and very reasonable. The theoretical results are immediately following the ideas. The algorithm proposed has a clear structure and is easy to understand and implement. For experiments, the new algorithm consistently outperforms existing studies on a set of MDPs where there exists a goal state (as both the unique reward state and the terminal state). Some important discussions are highlighted to introduce the algorithm,. Moreover, the proposed mechanism seems to be an inspiration for future work considering the state space exploration related to sample efficiency.

However, I wasn't fully convinced by the paper about relevance. Reinforcement learning aims to learn in an environment by trial and error without prior knowledge about the environment. As the problems considered by the paper are with episodic rewards (in both theory and experiments), *the problem themselves are shortest path problems*. Using the shortest path constraint to solve shortest path problems seems not fair to be placed among a set of learning algorithms. Armed with this prior knowledge, the algorithm outperforms marginally (though consistently) compared with pure learning-based algorithms, only with its best choice of k. I believe it would fall short if placed among search algorithms. Some strong justifications are needed for the work to be relevant.

Pros:
1.	The paper considers a practical problem in reinforcement learning: sample efficiency in sparse reward tasks. The RL algorithm tends to fail if the collected trajectory does not contain enough evaluative feedback. The idea of using a constrained-RL framework and cost function to tackle the problem is natural and has been well motivated given some drawbacks in existing work mentioned in section 5.
2.	The relaxation from the shortest path to k-SP is well explained. The novel cost function introduced to penalizes the policy-violating SP constraint can tackle some limitations of existing methods. For example, it can preserve the convergence and optimality of the policy.
3.  This paper provides convincing numerical experiments to show the effectiveness of the proposed framework. The ablation studies are also helpful to show the effects of hyperparameters.

Cons:
1. The choice of k is not clear.

---

> ### Author Response · Authors · 2020-11-18
> **Clarification on the use of prior knowledge, empirical performance, and the choice of $k$ of SPRL + comparison with search algorithms**
>
> We appreciate AnonReviewer4 for providing constructive feedback. Please refer to the common response above as well as the individual responses below.
>
> **Q**: “SPRL uses the prior knowledge of shortest path constraint.”
>
> **A**: Our SPRL does **not** exploit/require any extra knowledge other than the input observation. The shortest-path constraint is implemented using the RNet, and RNet is trained purely from the agent’s experience; thus, it is no different setup than other (general) intrinsically motivated exploration methods such as ICM.
>
> **Q**: “SPRL only outperforms marginally with the pure learning-based algorithms.”
>
> **A**: We argue that SPRL significantly outperforms all the pure learning-based algorithms on both Minigrid and DMlab. The GT-Grid agent, the second-best performing method in most of the tasks, is **not** a pure learning-based algorithm. In fact, GT-Grid uses the ground-truth state visitation count information, which is not available to other compared methods including SPRL. The main purpose of GT-Grid is to simulate the **upper-bound** performance of novelty-seeking methods (such as ICM, ECO, RND), and show that SPRL can even outperform them. Once we exclude GT-Grid, we can see that the performance gap between SPRL and other baselines are quite significant.
>
> **Q**: “Comparison of SPRL with search algorithms.”
>
> **A**: Since SPRL does not use a model or any form of prior knowledge, it is not fair to compare with search algorithms which requires the model (e.g., MCTS [Peter et al., 2002], etc) or the assumption of single-goal MDP (e.g., graph-based planning [Huang et al., 2019; Laskin et al. 2020; Savinov et al., 2018], etc). Instead we compared with the model-free method (PPO) and strong exploration methods designed specifically for sparse reward tasks (ICM, ECO), and the upper-bound performance of novelty-seeking exploration method (GT-Grid).
>
> **Q**: “The choice of $k$ is not clear.”
>
> **A**: As shown in Appendix A.1, we searched for the best $k$ since there exists a “sweet spot” for the choice of $k$; the SPRL cost becomes too sparse if $k$ is too large, and we get less reduction in trajectory space if $k$ is too small as suggested by the Lemma 2.
>
> [1] Peter Auer, Nicolo Cesa-Bianchi, and Paul Fischer. “Finite-time analysis of the multiarmed bandit problem.” Machine learning, 2002.
> [2] Zhiao Huang, Fangchen Liu, and Hao Su. “Mapping state space using landmarks for universal goal reaching.” In NeurIPS, pp. 1940–1950, 2019.
> [3] Michael Laskin, Scott Emmons, Ajay Jain, Thanard Kurutach, Pieter Abbeel, and Deepak Pathak. “Sparse graphical memory for robust planning.” arXiv preprint. arXiv:2003.06417, 2020.
> [4] Savinov, Nikolay, Alexey Dosovitskiy, and Vladlen Koltun." Semi-parametric topological memory for navigation." ICLR, 2018.

---

> > ### Comment · AnonReviewer4 · 2020-11-20
> > **Further discussions**
> >
> > I thank the author for your detailed response.
> >
> > Q: “SPRL uses the prior knowledge of shortest path constraint.”
> > Q: “SPRL only outperforms marginally with the pure learning-based algorithms.”
> >
> > Indeed SPRL does not have access to the model, but by applying this shortest path constraint the algorithm is putting *some* prior on the model of the game. When such a prior is accurate, it benefits; when such a prior is biased, it hurts. This is not model-based but not either completely model-free. It's somewhere in between indirect reinforcement learning and model-free RL. Therefore, when you put constraints like this, I'm focusing on the tradeoff between how general this constraint is to cover a large enough subset of problems of interest and how much performance improvements you gain for this subset of problems.
> >
> > For the former, your cover MDPs with a unique goal. This goal is the unique reward state and the unique terminal state. It is very general. As you mentioned in your response, some other tasks like ObjectMany can also be solved despite not rigorously lying in this subset. It is a good demonstration of the generalizability but naturally I would like to know how SPRL performs in an even wider set of problems. This will be an important source of information to determine it.
> >
> > For the latter, you're right that comparing to search is a bit too harsh for SPRL. But I'm expecting something beyond pure learning as I wouldn't be surprised if SPRL just outperforms pure learning-based methods in these tasks. GT-grid can be a good baseline, but it's the only one with a bit assumption of the world. It's might not be sufficient to fully determine the strength of SPRL, even under the subset of problems you present.
> >
> > I would be happy to hear the opposite idea, though.

---

> > > ### Author Response · Authors · 2020-11-21
> > > **Clarification on the prior used in SPRL compared to GT-Grid, ICM, ECO, PPO + entropy regularization**
> > >
> > > We appreciate the quick follow-up. We would like to clarify several points on SPRL’s prior compared to other baseline methods (ICM, ECO, PPO+entropy regularization).
> > >
> > > > “Yours cover MDPs with a unique goal” +  “ObjectMany… not rigorously lying in this subset”
> > >
> > > We note that both our theory and the resulting practical algorithm do **not** assume a unique goal; in fact, the *only* assumption we make on MDP is the mild stochasticity (See Theorem 3). Thus, even though the ObjectMany task has multiple goal states, since it satisfies the mild stochasticity assumption, it lies in our MDP of interest. To be specific, we extend the scope from single-goal MDP to general (multi-goal) MDP by extending Definition 1 to Definition 2.
> > >
> > > > “By applying SP-constraint it is putting some prior on the model of the game. When such a prior is accurate, it benefits; when such a prior is biased, it hurts.”
> > >
> > > It is indeed true; any form of regularization or constraint will change the solution space in a certain way, as many intrinsic exploration methods often do. We will clarify the trade-off that SPRL makes on given MDP. The **only** assumption SPRL makes on MDP is the mild stochasticity and as long as such an assumption is satisfied, SPRL is **guaranteed to converge to optimal policy**.  Moreover, Lemma 2 shows that SPRL can improve sample efficiency by reducing the policy search space. In the experiment, we showed that our SPRL provides even larger empirical benefits than other novelty-seeking methods. Lastly, we note that SPRL is expected to be more helpful in sparse-reward tasks; even in dense-reward tasks, it is guaranteed not to be worse than the base RL method (e.g. PPO in our experiments). Thus, we claim that SPRL makes a quite general assumption such that it can be used for a wide range of problems with strong empirical benefit.
> > >
> > > > “GT-Grid is the only one with a bit assumption of the world”
> > >
> > > We argue that GT-Grid is using much stronger prior (environment-specific information) than other methods including SPRL. SPRL, ECO, ICM, and PPO learn purely from the agent’s trajectory while GT-Grid uses an **environment-specific knowledge** which is the ground-truth state visitation count, not available to the agent. Using this strong prior, GT-Grid implements an optimal exploration strategy (i.e., UCB) and it simulates the upper-bound performance of the novel-seeking exploration methods. Moreover, the extent to which SPRL is taking advantage of prior knowledge is not significantly greater than other novelty-based exploration methods are.

---

> > > > ### Comment · AnonReviewer4 · 2020-11-21
> > > > **Further discussions**
> > > >
> > > > Indeed mild stochasticity, a weaker version of goal-directed MDP, is the assumption. It is again a general enough subset of MDP problems to study though not universal. For experiments indeed GT-Grid is very specific but the work still lacks some comparison on methods with weaker assumptions on the world. I believe the author and I have made a consensus on these points.
> > > >
> > > > I believe that the manuscript isn't perfect but it has some value for publicity at least for its employment of RNet and some applicability on mild stochasticity MDP. I now vote, very weakly, for acceptance.

---

### Official Review · AnonReviewer1 · 2020-10-29
**An interesting approach but might need to consider more diversified domains to justify its general applicability.**

**Rating:** 6
**Confidence:** 3

**Review:**

This paper proposed a k-shortest-path constrained reinforcement learning method for solving sparse reward MDPs. Under the assumption that the MDP is a single-goal task and by utilizing a novel cost function, the proposed method is demonstrated to outperform a few baselines in terms of sample efficiency.

Pro:
The technique along with its theoretical property are well developed.

Careful numerical studies are provided to evaluate the effect of the k-shortest constraint in a tabular-RL settings

Cons:
It is unclear how this method and its theoretical results can be generalized to other settings where the rewards are less sparse than single-goal MDP settings, but still very sparse.

The experiments for testing the proposed method are all based on navigation tasks, which give us an impression that the proposed method is only applicable to a specific problem. It might be better to consider a different type of domains.

The compared baseline methods are very limited and not necessarily ideal candidates for solving sparse reward problems. There are other approaches, such as hierarchical reinforcement learning which can also solve sparse reward problems , e.g., eigen-options [1] might also worth to compare.
[1] Machado et al. A Laplacian Framework for Option Discovery in Reinforcement Learning, imcl2017.

Other questions:
Is there a convergence guarantee for algorithm 1?

It might be better to provide some computational complexity analysis of algorithm 1.

What is the motivation of using k-reachability network to implement the binary distance discriminator?

--------------after the rebuttal---------------
I appreciated the authors' effort in addressing my comments and questions. I maintain my score of weak acceptance for this paper.

---

> ### Author Response · Authors · 2020-11-18
> **Clarification on the convergence guarantee, computational complexity and motivation of using RNet**
>
> We appreciate AnonReviewer1 for providing constructive feedback. Please refer to the common response above as well as the individual responses below.
>
> **Q:** “Does algorithm 1 has a convergence guarantee?”
>
> **A:** With the fixed (or ground-truth) RNet, SPRL has a convergence guarantee from the policy gradient theorem. Thus, strictly speaking, we should freeze the RNet after it is trained enough (e.g., after 50% of the training budget) to ensure convergence of policy. However, in practice, we found that RNet quickly converges as shown in Figure 13 and 14. Thus, we did not freeze the RNet weight in the experiments.
>
> **Q:** “Computational complexity analysis of SPRL”
>
> **A:** Here is the summary of computational complexity comparison of PPO and SPRL (excluding the complexity of PPO part).
>
> SPRL: 1 forward and 5 backward propagation of RNet / environment step
> PPO: 1 forward and 4 backward propagation of policy network / environment step
>
> Details: Compared to PPO, our SPRL additionally performs 1) offline training of RNet and 2) computation of SPRL cost term in Eq.(13). RNet takes two observations of size $S$ as input, and we trained RNet every $T$ environment steps over the batch of data of size B sampled from the replay buffer; thus, the computational complexity per environment step is $O(SB/T)$.  In our experiment, we used $B=30,000$ and $T=6,000$, so we performed an average 5 times of back-propagation of the RNet per environment step. We compute the SPRL cost every environment step, and in Eq.(13), the complexity of first term is $O(S)$ and second term is $O(k+\Delta t)$ and the third term is $O(1)$; thus, the overall complexity per environment step is $O(S)$ since $S >> k+ \Delta t$ on all the domains in our experiment. In other words, we perform a single pass of feed-forward of the RNet per environment step. PPO performs a single feed-forward pass of the policy network per environment step for rollout, and runs 4 iterations of update over the batch of experience (i.e., 4 times of backpropagation of the policy network per environment step).
>
> **Q:** “Motivation of using reachability network to implement binary distance discriminator”
>
> **A:** First of all, we note that the binary distance discriminator derived in Eq. (8) is **exactly the same** as the definition of reachability network; thus, it was a natural choice for us to use RNet. Another candidate we had considered is to learn a universal distance predictor similar to [Huang et al., 2019] and then threshold the predicted distance. However, since the problem of learning the universal distance predictor is **strictly harder** than learning the binary distance discriminator (i.e., optimization problem V.S. decision problem), we have chosen RNet. Please refer to the Section 5 “Distance metric…” paragraph for more discussion on other works utilizing the distance in MDP.
>
> [1] Zhiao Huang, Fangchen Liu, and Hao Su. “Mapping state space using landmarks for universal goal reaching.” In NeurIPS, pp. 1940–1950, 2019.

---

### Official Review · AnonReviewer3 · 2020-10-29
**A well written paper introducing shortest path constraint for sparse reward MDPs**

**Rating:** 6
**Confidence:** 4

**Review:**

**Summary**
This paper proposes a new constraint for constrained MDP, based on k-shortest path, which helps improve sample efficiency for (model-free) RL algorithms in sparse-reward MDP, while theoretically proving that the constraint retains the same optimal policy in the original MDP. Intuitively, for sparse (positive) reward setting, the optimal policy should reach the positive reward states with the shortest path (as it has the lowest discounting). The relaxed form of the constraint considers the that the distance between two states is less than $k$, rather than being optimal length (which the optimal policy still also satisfies). The constraint is then converted into its Lagrangian form as a cost term to the reward (i.e. a type of reward shaping). Practically, this requires a k-reachability network (RNet), which is a binary distance discriminator judging whether the distance between two states are reachable within $k$ steps. This network is trained with contrastive loss, similar to prior work SPTM by Savinov et al., 2018. However, SPTM uses RNet for graph-based planning (i.e. the local distance between states), while this paper uses RNet as the cost/constraint on the policy objective function. Experiments were conducted in several maze navigation environments (2D grid world MiniGrid, to first person 3D maze environments in DeepMind Lab), showing promising results compared to several baselines which use intrinsic curiosity. Several ablations were performed on the hyperparameters ($k$, and tolerance $\delta t$ on the constraint), as well as some qualitative examples of the policies learned compared to novelty reward shaping.


**Strengths**:
- There has been a lot of work with goal-conditioned RL with sparse reward. The advantage of this paper is that this approach converts the notion of reaching goals into a constraint that is applicable to general MDPs.
- The theoretical results (under some mild assumptions such as positive cumulative reward, mild stochasticity) appear to be correct, although I am not absolutely certain.
- Empirically the experiments were designed well to study the effects of its hyperparameters ($k$, and tolerance $\delta t$ on the constraint), and performs strongly as the task is harder (sparser reward / long trajectories) compared to its baselines

**Weaknesses**:
- It is still unclear to me about the potential performance gap between using an imperfect RNet versus an oracle ground truth distance discriminator. See my question below for some more detail
- In the experimental sections, some sections are unclear about whether the RNet is used or not. For example, RNet is not used in 6.2 (MiniGrid), while is used in 6.3 (DeepMindLab) but required reading appendix to confirm. I am not sure for 6.4 and 6.5 whether RNet is used at all (I think it is not). Please clarify for me and in the paper.

**Recommendation**:
 I recommend this paper a marginally above acceptance threshold. Overall I think that it is a well-written paper with thorough theoretical and empirical results. There are some clarification parts that can improve the paper even further.

**Questions**:
1. My main question is what is the performance difference between having the ground truth distance versus using the RNet? For example, can you compare the performance of 6.2 if RNet was used here? I understand RNet is not used in 6.2 for the purpose of understanding the objective. I would the oracle RNet would give the upper bound on the performance on SPRL, but how much worse is using RNet in those environments?
2. Perhaps outside of the scope of this paper, but I am curious about how SPRL would perform if there was a curriculum on the $k$ and $\delta t$ value, rather than a fixed value that is treated as a hyper parameter. Perhaps the authors have some intuition or have actually tried this as well.
3. Please clarify about the use of RNet vs. Ground truth distance in the section 6.4/6.5.

**After rebuttal responses**:

I have read the authors’ response to my concerns, as well as the other reviews. I maintain my current evaluation with a weak acceptance of the paper.

---

> ### Author Response · Authors · 2020-11-18
> **Added more ablation studies and improved the clarity of our paper regarding the use of GT-RNet**
>
> We appreciate AnonReviewer3 for providing constructive feedback. Please refer to the common response above as well as the individual responses below.
>
> **Q**: “Comparison between SPRL with the learned RNet versus GT-RNet”
>
> **A**:  We compared the performance of SPRL with learned RNet and GT-RNet on Minigrid in Appendix B.2 . The Figure 15 shows that the SPRL with learned RNet performs similar to the one with GT-RNet. This is partly because the RNet can be trained quite quickly as shown in Appendix B.1. Also, we claim that a small noise in RNet output can be even helpful since it has an effect of smoothing out the SPRL cost; i.e., instead of 0/1 discrete cost, the noisy cost can be seen as a real-numbered cost in [0, 1] in expectation, which works as a denser supervision for learning.
>
> **Q**: “It is unclear whether the (GT-)RNet is used or not”
>
> **A**:  Based on the updated draft, empirical evaluation in section 6.2 and 6.3 uses the learned RNet, and the analysis in 6.4 and 6.5 are using GT-RNet. We have made it more clear in the updated paper as follows:
> * Section 6.1: “Following Savinov et al. (2018), we trained RNet ... on both MiniGrid (Section 6.2) and DeepMind Lab (Section 6.3).”
> * Section 6.4: “We used the ground-truth distance function instead of the learned RNet for the exact analysis.”
> * Section 6.5: “We qualitatively study … k-SP constraint with the ground-truth RNet”
>
> **Q**: “How SPRL would perform with a curriculum on k and $\Delta t$ (i.e., tolerance)”
>
> **A**:  We evaluated SPRL with a curriculum on $k$. The result on MiniGrid is summarized in Figure 16 in Appendix B.3. The result shows that curriculum learning does not improve the performance, and even hurts the performance for KeyDoor-7x7 and KeyDoor-11x11 tasks. In fact, curriculum learning on k can make the policy learning unstable, since the reward function of MDP changes (i.e., non-stationary) when $k$ changes. Thus, we did not use the curriculum learning for all the experiments.
>
> [1] Nikolay Savinov, Anton Raichuk, Rapha ̈el Marinier, Damien Vincent, Marc Pollefeys, Timothy Lillicrap, and Sylvain Gelly. "Episodic curiosity through reachability." ICLR, 2018

---

> > ### Author Response · Authors · 2020-11-19
> > **Added curriculum learning result on DeepMind Lab**
> >
> > We added the curriculum learning experiment result on DeepMind Lab in addition to the previous result on MiniGrid in Appendix B.3. The DeepMind Lab result also indicates that adding a curriculum on $k$ makes the RNet training slightly more unstable, and the performance of SPRL slightly worse.

---

> > > ### Comment · AnonReviewer3 · 2020-11-21
> > > **Thanks for addressing my concerns**
> > >
> > > Thank you to the authors for addressing my questions and running the requested baseline experiments. The updated draft is much more clear now in terms of the use of learned vs ground truth RNet.
> > >
> > > The experiments in B.1 are interesting to show that even with around 80-90% accuracy in the RNet, the performance of SPRL is almost similar to using the ground-truth, possibly due to the noise in RNet output. This is not required for the rebuttal but it may be worth investigating what is the confusion matrix of RNet over time (i.e. is it predicting 1 when actually 0 more often or predicting 0 when it is 1).
> > >
> > > It is also great that you have tried the experiment on the curriculum on $k$ which appears to not help (hypothesized as training instability due to shifting targets). I also appreciate that the changes in the draft are colour coded.

---

> > > > ### Author Response · Authors · 2020-11-22
> > > > **Confusion matrix of RNet**
> > > >
> > > > Thank you for recognizing the improvements we made in the paper. We calculated the confusion matrix of RNet on MiniGrid tasks after training is over. The result shows that the RNet is not particularly biased, and mostly makes the correct prediction.
> > > >
> > > > * FourRoom - 7x7 @ 0.5M steps
> > > >
> > > > |           | Actual 1 | Actual 0 |
> > > > |-----------|----------|----------|
> > > > | **Predict 1** | 12.5K     | 2.5K      |
> > > > | **Predict 0** | 1.5K      | 13.5K       |
> > > >
> > > > * KeyDoor - 7x7 @ 1M steps
> > > >
> > > > |           | Actual 1 | Actual 0 |
> > > > |-----------|----------|----------|
> > > > | **Predict 1** | 14.5K  | 0.5K     |
> > > > | **Predict 0** | 0.5K    | 14.5K   |

---

### Author Response · Authors · 2020-11-18
**Common responses + summary of the updates in the paper**

We truly appreciate all the constructive comments by the reviewers. All reviewers recognized that our paper is well-written, the idea is novel and intuitive, and our theoretical result is sound. Also, our work presents the well-structured experiments (R1, R3) with strong empirical results (R2, R4). We will reflect all the comments from reviewers in the revision.

**Updates in the paper**

We updated the paper to reflect reviewers' comments and to improve clarity. The changed texts are colored in red in the updated draft. Here are the summary of major changes:
* Section 6.2 - Figure 3: We noticed that using GT-RNet in Minigrid caused a confusion that SPRL requires the GT-RNet, which is not true. To avoid such a confusion, we re-ran the MiniGrid experiment with the learned RNet and updated the Figure 3 in the paper.
* Section 6: We made it more explicit that the GT-RNet was used only for the analysis (i.e., Section 6.4 and 6.5)
* Appendix B: We added a new section providing more details on the learned RNet’s accuracy and a new experiment result comparing our SPRL with learned and GT-RNet.

**Common responses**

**R1+R4** : “The proposed method assumes/was evaluated on the single-goal (or goal-conditioned) MDP”

Our method does **not**assume a single-goal MDP; in fact, it is one of our main contributions to extend the notion of shortest-path from single-goal MDP to the multi-goal and/or non-deterministic MDP. For example, the ObjectMany task has multiple rewarding items on the map and the agent should collect them as many as possible in the limited time, and our SPRL achieves a high performance on this task as shown in Figure 4.

**R1+R2**: “Comparison to HRL algorithms (e.g., HIRO, HER)”

HRL algorithms such as HIRO [Nachum et al., 2018] and HER [Andrychowicz et al., 2017] assume the single-goal task and availability to the goal-distance function. One of our main contributions is that our method can be used without such assumptions, and both DMLab and Minigrid are not in a goal-conditioned RL setting (i.e., do not provide the agent with the goal location and the goal distance function). Thus, we instead compared with the exploration methods developed for a generic (sparse-reward) MDP without any additional assumptions.
Also, we emphasize that GT-Grid is a very strong baseline, and our SPRL consistently outperforms GT-Grid. GT-Grid simulates the **upper-bound performance**of the novelty-based exploration methods (e.g., ICM, RND [Burda et al., 2019], ECO, etc) by using the ground-truth state visitation count information.

**R1+R2**: “Would SPRL scale to the non-navigational tasks with sparser reward and larger state space?”

We evaluated SPRL with the learned RNet on the non-navigational task with larger state space: Montezuma’s Revenge. We ran the experiment for only 10M environment steps (or 40M frames) and searched over a small set of hyperparameters, given the short rebuttal period. We compare SPRL with CoEX and RND in terms of how many environment steps are required to solve the first room (i.e., score of 400).
* SPRL: 400 @ 5M environment steps (averaged over 3 seeds)
* CoEX: 400 @ 10M environment steps (from Figure 5 in [Choi et al., 2019])
* RND: 400 @ 8M environment steps (from Figure 6 in [Burda et al., 2019])

We can see that SPRL can quickly learn to solve the first room, which is comparable to SOTA exploration methods such as CoEX [Choi et al., 2019] and RND [Burda et al., 2019].

[1] Nachum et al. "Data-efficient hierarchical reinforcement learning." Advances in Neural Information Processing Systems. 2018.
[2] Andrychowicz et al. "Hindsight experience replay." Advances in Neural Information Processing Systems. 2017.
[3] Choi et al. "Contingency-Aware Exploration in Reinforcement Learning." ICLR. 2019.
[4] Burda et al. "Exploration by random network distillation." ICLR. 2019.

---

### Decision · Program_Chairs · 2021-01-07
**Final Decision**

**Decision:**

Reject

**Comment:**

This work proposes a shortest path constraint for the reinforcement learning algorithm to improve efficiency in sparse-reward scenarios. The experiments are shown in navigation tasks in first-person maze and grid world. Reviewers found the idea interesting and the paper well-written but none of them championed the paper for clear acceptance. The authors provided a detailed thoughtful rebuttal. All the reviewers acknowledged the rebuttal followed by discussion. After considering rebuttal, review, and discussion, both AC and reviewers feel that experiments don't fully support and justify the algorithm. The main issue is that the results are shown only for the shortest pathfinding problems where the shortest path constraint is shown to be helpful. Hence, it is recommended to run it on diverse scenarios and standard benchmarks like the Atari games suite. Please refer to the reviews for final feedback and suggestions to strengthen the future submission.